# Adversarial Robustness of Supervised Sparse Coding

**Jeremias Sulam**
Johns Hopkins University
`jsulam1@jhu.edu`

**Ramchandran Muthukumar**
Johns Hopkins University
`rmuthuk1@jhu.edu`

**Raman Arora**
Johns Hopkins University
arora@cs.jhu.edu

## Abstract

Several recent results provide theoretical insights into the phenomena of adversarial examples. Existing results, however, are often limited due to a gap between the simplicity of the models studied and the complexity of those deployed in practice. In this work, we strike a better balance by considering a model that involves learning a representation while at the same time giving a precise generalization bound and a robustness certificate. We focus on the hypothesis class obtained by combining a sparsity-promoting encoder coupled with a linear classifier, and show an interesting interplay between the expressivity and stability of the (supervised) representation map and a notion of margin in the feature space. We bound the robust risk (to $\ell_2$-bounded perturbations) of hypotheses parameterized by dictionaries that achieve a mild encoder gap on training data. Furthermore, we provide a robustness certificate for end-to-end classification. We demonstrate the applicability of our analysis by computing certified accuracy on real data, and compare with other alternatives for certified robustness.

## 1 Introduction

With machine learning applications becoming ubiquitous in modern-day life, there exists an increasing concern about the robustness of the deployed models. Since first reported in [Szegedy et al., 2013, Goodfellow et al., 2014, Biggio et al., 2013], these *adversarial attacks* are small perturbations of the input, imperceptible to the human eye, which can nonetheless completely fluster otherwise well-performing systems. Because of clear security implications [DARPA, 2019], this phenomenon has sparked an increasing amount of work dedicated to devising defense strategies [Metzen et al., 2017, Gu and Rigazio, 2014, Madry et al., 2017] and correspondingly more sophisticated attacks [Carlini and Wagner, 2017, Athalye et al., 2018, Tramer et al., 2020], with each group trying to triumph over the other in an arms-race of sorts.

A different line of research attempts to understand adversarial examples from a theoretical standpoint. Some works have focused on giving robustness certificates, thus providing a guarantee to withstand the attack of an adversary under certain assumptions [Cohen et al., 2019, Raghunathan et al., 2018, Wong and Kolter, 2017]. Other works address questions of learnabiltiy [Shafahi et al., 2018, Cullina et al., 2018, Bubeck et al., 2018, Tsipras et al., 2018] or sample complexity [Schmidt et al., 2018, Yin et al., 2018, Tu et al., 2019], in the hope of better characterizing the increased difficulty of learning hypotheses that are robust to adversarial attacks. While many of these results are promising, the analysis is often limited to simple models.

Here, we strike a better balance by considering a model that involves learning a representation while at the same time giving a precise generalization bound and a robustness certificate. In particular, we focus our attention on the adversarial robustness of the supervised sparse coding model [Mairal et al., 2011], or task-driven dictionary learning, consisting of a linear classifier acting on the representation computed via a supervised sparse encoder. We show an interesting interplay between the expressivity and stability of a (supervised) representation map and a notion of margin in the feature space. The idea of employing sparse representations as data-driven features for supervised learning goes back to the early days of deep learning [Coates and Ng, 2011, Kavukcuoglu et al., 2010, Zeiler et al., 2010, Ranzato et al., 2007], and has had a significant impact on applications in computer vision and machine learning [Wright et al., 2010, Henaff et al., 2011, Mairal et al., 2008, 2007, Gu et al., 2014]. More recently, new connections between deep networks and sparse representations were formalized by Papyan et al. [2018], which further helped deriving stability guarantees [Papyan et al., 2017b], providing architecture search strategies and analysis [Tolooshams et al., 2019, Murdock and Lucey, 2020, Sulam et al., 2019], and other theoretical insights [Xin et al., 2016, Aberdam et al., 2019, Aghasi et al., 2020, Aberdam et al., 2020, Moreau and Bruna, 2016]. While some recent work has leveraged the stability properties of these latent representations to provide robustness guarantees against adversarial attacks [Romano et al., 2019], these rely on rather stringent generative model assumptions that are difficult to be satisfied and verified in practice. In contrast, our assumptions rely on the existence of a positive *gap* in the encoded features, as proposed originally by Mehta and Gray [2013]. This distributional assumption is significantly milder – it is directly satisfied by making traditional sparse generative model assumptions – and can be directly quantified from data.

This work makes two main contributions: The first is a bound on the robust risk of hypotheses that achieve a mild encoder gap assumption, where the adversarial corruptions are bounded in $\ell_2$-norm. Our proof technique follows a standard argument based on a minimal $\epsilon$-cover of the parameter space, dating back to Vapnik and Chervonenkis [1971] and adapted for matrix factorization and dictionary learning problems in Gribonval et al. [2015]. However, the analysis of the Lipschitz continuity of the adversarial loss with respect to the model parameters is considerably more involved. The increase in the sample complexity is mild with adversarial corruptions of size $\nu$ manifesting as an additional term of order $\mathcal{O}\left((1+\nu)^2/m\right)$ in the bound, where $m$ is the number of samples, and a minimal encoder gap of $\mathcal{O}(\nu)$ is necessary. Much of our results extend directly to other supervised learning problems (e.g. regression). Our second contribution is a robustness certificate that holds for every hypothesis in the function class for $\ell_2$ perturbations for multiclass classification. In a nutshell, this result guarantees that the label produced by the hypothesis will not change if the encoder gap is *large enough* relative to the energy of the adversary, the classifier margin, and properties of the model (e.g. dictionary incoherence).

## 2 Preliminaries and Background

In this section, we first describe our notation and the learning problem, and then proceed to situate our contribution in relation to prior work.

Consider the spaces of inputs, $\mathcal{X} \subseteq B_{\mathbb{R}^d}$, i.e. the unit ball in $\mathbb{R}^d$, and labels, $\mathcal{Y}$. Much of our analysis is applicable to a broad class of label spaces, but we will focus on binary and multi-class classification setting in particular. We assume that the data is sampled according to some unknown distribution $P$ over $\mathcal{X} \times \mathcal{Y}$. Let $\mathcal{H} = \{f : \mathcal{X} \to \mathcal{Y}'\}$ denote a hypothesis class mapping inputs into some output space $\mathcal{Y}' \subseteq \mathbb{R}$. Of particular interest to us are norm-bounded linear predictors, $f(\cdot) = \langle \mathbf{w}, \cdot \rangle$, parametrized by $d$-dimensional vectors $\mathbf{w} \in \mathcal{W} = \{\mathbf{w} \in \mathbb{R}^d : \|\mathbf{w}\|_2 \leq B\}$.

From a learning perspective, we have a considerable understanding of the linear hypothesis class, both in a stochastic non-adversarial setting as well as in an adversarial context [Charles et al., 2019, Li et al., 2019]. However, from an application standpoint, linear predictors are often too limited, and rarely applied directly on input features. Instead, most state-of-the-art systems involve learning a representation. In general, an *encoder* map $\varphi : \mathcal{X} \to \mathcal{Z} \subseteq \mathbb{R}^p$, parameterized by parameters $\theta$, is composed with a linear function so that $f(\mathbf{x}) = \langle \mathbf{w}, \varphi_\theta(\mathbf{x}) \rangle$, for $\mathbf{w} \in \mathbb{R}^p$. This description applies to a large variety of popular models, including kernel-methods, multilayer perceptrons and deep convolutional neural networks. Herein we focus on an encoder given as the solution to a Lasso problem [Tibshirani, 1996]. More precisely, we consider $\varphi_\mathbf{D}(\mathbf{x}) : \mathbb{R}^d \to \mathbb{R}^p$, defined by

$$\varphi_\mathbf{D}(\mathbf{x}) \coloneqq \arg\min_\mathbf{z} \frac{1}{2}\|\mathbf{x} - \mathbf{D}\mathbf{z}\|_2^2 + \lambda\|\mathbf{z}\|_1. \tag{1}$$

Note that when $\mathbf{D}$ is overcomplete, i.e. $p > d$, this problem is not strongly convex. Nonetheless, we will assume that that solution to Problem 1 is unique[1], and study the hypothesis class given by $\mathcal{H} = \{f_{\mathbf{D},\mathbf{w}}(\mathbf{x}) = \langle \mathbf{w}, \varphi_{\mathbf{D}}(\mathbf{x}) \rangle : \mathbf{w} \in \mathcal{W}, \mathbf{D} \in \mathcal{D}\}$, where $\mathcal{W} = \{\mathbf{w} \in \mathbb{R}^p : \|\mathbf{w}\|_2 \leq B\}$, and $\mathcal{D}$ is the oblique manifold of all matrices with unit-norm columns (or *atoms*); i.e. $\mathcal{D} = \{\mathbf{D} \in \mathbb{R}^{d \times p} : \|\mathbf{D}_i\|_2 = 1 \; \forall i \in [p]\}$. While not explicit in our notation, $\varphi_{\mathbf{D}}(\mathbf{x})$ depends on the value of $\lambda$. For notational simplicity, we also suppress subscripts $(\mathbf{D}, \mathbf{w})$ in $f_{\mathbf{D},\mathbf{w}}(\cdot)$ and simply write $f(\cdot)$.

We consider a bounded loss function $\ell : \mathcal{Y} \times \mathcal{Y}' \to [0, b]$, with Lipschitz constant $L_\ell$. The goal of learning is to find an $f \in \mathcal{H}$ with minimal risk, or expected loss, $R(f) = \mathbb{E}_{(\mathbf{x},y) \sim P}[\ell(y, f(\mathbf{x}))]$. Given a sample $S = \{(\mathbf{x}_i, y_i)\}_{i=1}^m$, drawn i.i.d. from $P$, a popular learning algorithm is empirical risk minimization (ERM) which involves finding $f_{\mathbf{D},\mathbf{w}}$ that solves the following problem:

$$\min_{\mathbf{D},\mathbf{w}} \frac{1}{m} \sum_{i=1}^m \ell(y_i, f_{\mathbf{D},\mathbf{w}}(\mathbf{x}_i)).$$

**Adversarial Learning.** In an adversarial setting, we are interested in hypotheses that are robust to adversarial perturbations of inputs. We focus on *evasion attacks*, in which an attack is deployed at test time (while the training samples are not tampered with). As a result, a more appropriate loss that incorporates the robustness to such contamination is the robust loss [Madry et al., 2017], $\tilde{\ell}_\nu(y, f(\mathbf{x})) := \max_{\mathbf{v} \in \Delta_\nu} \ell(y, f(\mathbf{x} + \mathbf{v}))$, where $\Delta$ is some subset of $\mathbb{R}^d$ that restricts the power of the adversary. Herein we focus on $\ell_2$ norm-bounded corruptions, $\Delta_\nu = \{\mathbf{v} \in \mathbb{R}^d : \|\mathbf{v}\|_2 \leq \nu\}$, and denote by $\tilde{R}_S(f) = \frac{1}{m}\sum_{i=1}^m \tilde{\ell}_\nu(y_i, f(\mathbf{x}_i))$ the empirical robust risk of $f$ and $\tilde{R}(f) = \mathbb{E}_{(\mathbf{x},y) \sim P}[\tilde{\ell}_\nu(y, f(\mathbf{x}))]$ its population robust risk w.r.t. distribution $P$.

**Main Assumptions.** We make two general assumptions throughout this work. First, we assume that the dictionaries in $\mathcal{D}$ are $s$-incoherent, i.e, they satisfy a restricted isometry property (RIP). More precisely, for any $s$-sparse vector, $\mathbf{z} \in \mathbb{R}^p$ with $\|\mathbf{z}\|_0 = s$, there exists a minimal constant $\eta_s < 1$ so that $\mathbf{D}$ is close to an isometry, i.e. $(1 - \eta_s)\|\mathbf{z}\|_2^2 \leq \|\mathbf{D}\mathbf{z}\|_2^2 \leq (1 + \eta_s)\|\mathbf{z}\|_2^2$. Broad classes of matrices are known to satisfy this property (e.g. sub-Gaussian matrices [Foucart and Rauhut, 2017]), although empirically computing this constant for a fixed (deterministic) matrix is generally intractable. Nonetheless, this quantity can be upper bounded by the correlation between columns of $\mathbf{D}$, either via mutual coherence [Donoho and Elad, 2003] or the Babel function [Tropp et al., 2003], both easily computed in practice.

Second, we assume that the map $\varphi_{\mathbf{D}}$ induces a positive *encoder gap* on the computed features. Given a sample $\mathbf{x} \in \mathcal{X}$ and its encoding, $\varphi_{\mathbf{D}}(\mathbf{x})$, we denote by $\Lambda^{p-s}$ the set of atoms of cardinality $(p - s)$, i.e., $\Lambda^{p-s} = \{\mathcal{I} \subseteq \{1, \dots, p\} : |\mathcal{I}| = p - s\}$. The encoder gap $\tau_s(\cdot)$ induced by $\varphi_{\mathbf{D}}$ on any sample $\mathbf{x}$ is defined [Mehta and Gray, 2013] as

$$\tau_s(\mathbf{x}) := \max_{\mathcal{I} \in \Lambda^{p-s}} \min_{i \in \mathcal{I}} \left(\lambda - |\langle \mathbf{D}_i, \mathbf{x} - \mathbf{D}\varphi_{\mathbf{D}}(\mathbf{x})\rangle|\right).$$

An equivalent and conceptually simpler definition for $\tau_s(\mathbf{x})$ is the $(s+1)^{th}$ smallest entry in the vector $\lambda \mathbf{1} - |\langle \mathbf{D}, \mathbf{x} - \mathbf{D}\varphi_{\mathbf{D}}(\mathbf{x})\rangle|$. Intuitively, this quantity can be viewed as a measure of maximal energy along any dictionary atom that is not in the support of an input vector. More precisely, recall from the optimality conditions of Problem (1) that $|\mathbf{D}_i^T(\mathbf{x} - \mathbf{D}\varphi_{\mathbf{D}}(\mathbf{x}))| = \lambda$ if $[\varphi_{\mathbf{D}}(\mathbf{x})]_i \neq 0$, and $|\mathbf{D}_i^T(\mathbf{x} - \mathbf{D}\varphi_{\mathbf{D}}(\mathbf{x}))| \leq \lambda$ otherwise. Therefore, if $\tau_s$ is large, this indicates that there exist a set $\mathcal{I}$ of $(p - s)$ atoms that are *far* from entering the support of $\varphi_{\mathbf{D}}(\mathbf{x})$. If $\varphi_{\mathbf{D}}(\mathbf{x})$ has exactly $k$ non-zero entries, we may choose some $s > k$ to obtain $\tau_s(\mathbf{x})$. In general, $\tau_s(\cdot)$ depends on the energy of the residual, $\mathbf{x} - \mathbf{D}\varphi_{\mathbf{D}}(\mathbf{x})$, the correlation between the atoms, the parameter $\lambda$, and the cardinality $s$. In a nutshell, if a dictionary $\mathbf{D}$ provides a quickly decaying approximation error as a function of the cardinality $s$, then a positive encoder gap exists for some $s$.

We consider dictionaries that induce a positive encoder gap in every input sample from a dataset, and define the minimum such margin as $\tau_s^* := \min_{i \in [m]} \tau_s(\mathbf{x}_i) > 0$. Such a positive encoder exist for quite general distributions, such as $s$-sparse and approximately sparse signals. However, this definition is more general and it will allow us to avoid making any other stronger distributional assumptions. We now illustrate such the encoder gap with both analytic and numerical examples[2].

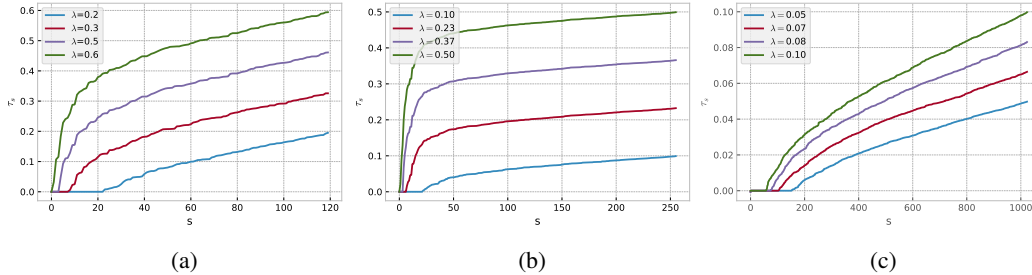

Figure 1: Encoder gap, $\tau_s^*$, for synthetic approximately sparse signals (a) MNIST digits (b) and CIFAR10 images (c).

**Approximate $k$-sparse signals** Consider signals $\mathbf{x}$ obtained as $\mathbf{x} = \mathbf{D}\mathbf{z} + \mathbf{v}$, where $\mathbf{D} \in \mathcal{D}$, $\|\mathbf{v}\|_2 \leq \nu$ and $\mathbf{z}$ is sampled from a distribution of sparse vectors with up to $k$ non-zeros, with $k < \frac{1}{3}\left(1 + \frac{1}{\mu(\mathbf{D})}\right)$, where $\mu(\mathbf{D}) = \max_{i \neq j}\langle \mathbf{D}_i, \mathbf{D}_j\rangle$ is the mutual coherence of $\mathbf{D}$. Then, for a particular choice of $\lambda$, we have that $\tau_s(\mathbf{x}) > \lambda - \frac{15\mu\nu}{2}, \forall s > k$. This can be shown using standard results in [Tropp, 2006]; we defer the proof to the Appendix A. Different values of $\lambda$ provide different values of $\tau_s(\mathbf{x})$. To illustrate this trade-off, we generate synthetic approximately $k$-sparse signals ($k = 15$) from a dictionary with 120 atoms in 100 dimensions and contaminate them with Gaussian noise. We then numerically compute the value of $\tau_s^*$ as a function of $s$ for different values of $\lambda$, and present the results in Figure 1a.

**Image data** We now demonstrate that a positive encoder exist for natural images as well. In Figure 1b we similarly depict the value of $\tau_s(\cdot)$, as a function of $s$, for an encoder computed on MNIST digits and CIFAR images (from a validation set) with learned dictionaries (further details in Section 6).

In summary, the encoder gap is a measure of the ability of a dictionary to sparsely represent data, and one can induce a larger encoder gap by increasing the regularization parameter or the cardinality $s$. As we will shortly see, this will provide us with a a controllable knob in our generalization bound.

## 3 Prior Work

Many results exist on the approximation power and stability of Lasso (see [Foucart and Rauhut, 2017]), which most commonly rely on assuming data is (approximately) $k$-sparse under a given dictionary. As explained in the previous section, we instead follow an analysis inspired by Mehta and Gray [2013], which relies on the encoder gap. Mehta and Gray [2013] leverage encoder gap to derive a generalization bound for the supervised sparse coding model in a stochastic (non-adversarial) setting. Their result, which follows a symmetrization technique [Mendelson and Philips, 2004], scales as $\tilde{\mathcal{O}}(\sqrt{(dp + \log(1/\delta))/m}$, and requires a minimal number of samples that is $\mathcal{O}(1/(\tau_s\lambda))$. In contrast, we study an generalization in the adversarially robust setting, detailed above. Our analysis is based on an $\epsilon$-cover of the parameter space and on analyzing a local-Lipschitz property of the adversarial loss. The proof of our generalization bound is simpler, and shows a mild deterioration of the upper bound on the generalization gap due to adversarial corruption.

Our work is also inspired by the line of work initiated by Papyan et al. [2017a] who regard the representations computed by neural networks as approximations for those computed by a Lasso encoder across different layers. In fact, a first analysis of adversarial robustness for such a model is presented by Romano et al. [2019]; however, they make strong generative model assumptions and thus their results are not applicable to real-data practical scenarios. Our robustness certificate mirrors the analysis from the former work, though leveraging a more general and new stability bound (Lemma 5.2) relying instead on the existence of positive encoder gap. In a related work, and in the context of neural networks, Cisse et al. [2017] propose a regularization term inspired by Parseval frames, with the empirical motivation of improving adversarial robustness. Their regularization term can in fact be related to minimizing the (average) mutual coherence of the dictionaries, which naturally arises as a control for the generalization gap in our analysis.

Lastly, several works have employed sparsity as a beneficial property in adversarial learning [Marzi et al., 2018, Demontis et al., 2016], with little or no theoretical analysis, or in different frameworks

(e.g. sparse weights in deep networks [Guo et al., 2018, Balda et al., 2019], or on different domains [Bafna et al., 2018]). Our setting is markedly different from that of Chen et al. [2013] who study adversarial robustness of Lasso as a sparse predictor directly on input features. In contrast, the model we study here employs Lasso as an encoder with a data-dependent dictionary, on which a linear hypothesis is applied. A few works have recently begun to analyze the effect of learned representations in an adversarial learning setting [Ilyas et al., 2019, Allen-Zhu and Li, 2020]. Adding to that line of work, our analysis demonstrates that benefits can be provided by exploiting a trade-off between expressivity and stability of the computed representations, and the classifier margin.

## 4 Generalization bound for robust risk

In this section, we present a bound on the robust risk for models satisfying a positive encoder gap. Recall that given a $b$-bounded loss $\ell$ with Lipschitz constant $L_\ell$, $\tilde{R}_S(f) = \frac{1}{m} \sum_{i=1}^m \tilde{\ell}_\nu(y_i, f(\mathbf{x}_i))$ is the empirical robust risk, and $\tilde{R}(f) = \mathbb{E}_{(\mathbf{x},y) \sim P}\big[\tilde{\ell}_\nu(y, f(\mathbf{x}))\big]$ is the population robust risk w.r.t. distribution $P$. Adversarial perturbations are bounded in $\ell_2$ norm by $\nu$. Our main result below guarantees that if a hypothesis $f_{\mathbf{D},\mathbf{w}}$ is found with a sufficiently large encoder gap, and a large enough training set, its generalization gap is bounded as $\tilde{\mathcal{O}}\big(b\sqrt{\frac{(d+1)p}{m}}\big)$, where $\tilde{\mathcal{O}}$ ignores poly-logarithmic factors.

**Theorem 4.1.** *Let $\mathcal{W} = \{\mathbf{w} \in \mathbb{R}^p : \|\mathbf{w}\|_2 \leq B\}$, and $\mathcal{D}$ be the set of column-normalized dictionaries with $p$ columns and with RIP at most $\eta_s^*$. Let $\mathcal{H} = \{f_{\mathbf{D},\mathbf{w}}(\mathbf{x}) = \langle \mathbf{w}, \varphi_{\mathbf{D}}(\mathbf{x})\rangle : \mathbf{w} \in \mathcal{W}, \mathbf{D} \in \mathcal{D}\}$. Denote $\tau_s^*$ the minimal encoder gap over the $m$ samples. Then, with probability at least $1 - \delta$ over the draw of the $m$ samples, the generalization gap for any hypothesis $f \in \mathcal{H}$ that achieves an encoder gap on the samples of $\tau_s^* > 2\nu$, satisfies*

$$\left|\tilde{R}_S(f) - \tilde{R}(f)\right| \leq \frac{b}{\sqrt{m}} \left((d+1)p\log\left(\frac{3m}{2\lambda(1-\eta_s^*)}\right) + p\log(B) + \log\frac{4}{\delta}\right)^{\frac{1}{2}}$$
$$+ b\sqrt{\frac{2\log(m/2) + 2\log(2/\delta)}{m}} + 12\frac{(1+\nu)^2 L_\ell B \sqrt{s}}{m},$$

*as long as $m > \frac{\lambda(1-\eta_s)}{(\tau_s^*-2\nu)^2} K_\lambda$, where $K_\lambda = \left(2\left(1 + \frac{1+\nu}{2\lambda}\right) + \frac{5(1+\nu)}{\sqrt{\lambda}}\right)^2$.*

A few remarks are in order. First, note that adversarial generalization incurs a polynomial dependence on the adversarial perturbation $\nu$. This is mild, especially since it only affects the fast $\mathcal{O}(1/m)$ term. Second, the bound requires a minimal number of samples. Such a requirement is intrinsic to the stability of Lasso (see Lemma 4.2 below) and it exists also in the non-adversarial setting [Mehta and Gray, 2013]. In the adversarial case, this requirement becomes more demanding, as reflected by the term $(\tau_s^* - 2\nu)$ in the denominator. Moreover, a minimal encoder gap $\tau_s^* > 2\nu$ is needed as well.

Theorem 4.1 suggests an interesting trade-off. One can obtain a large $\tau_s^*$ by increasing $\lambda$ and $s$ – as demonstrated in in Figure 1. But increasing $\lambda$ may come at an expense of hurting the empirical error, while increasing $s$ makes the term $1 - \eta_s$ smaller. Therefore, if one obtains a model with small training error, along with large $\tau_s^*$ over the training samples for an appropriate choice of $\lambda$ and $s$ while ensuring that $\eta_s$ is bounded away from 1, then $f_{\mathbf{D},\mathbf{w}}$ is guaranteed to generalize well. Furthermore, note that the excess error depends mildly (poly logarithmically) on $\lambda$ and $\eta_s$.

Our proof technique is based on a minimal $\epsilon$-cover of the parameter space, and the full proof is included in the Appendix B. Special care is needed to ensure that the encoder gap of the dictionary holds for a sample drawn from the population, as we can only measure this gap on the provided $m$ samples. To address this, we split the data equally into a training set and a development set: the former is used to learn the dictionary, and the latter to provide a high probability bound on the event that $\tau_s(\mathbf{x}) > \tau_s^*$. This is to ensure that the random samples of the encoder margin are i.i.d. for measure concentration. Ideally, we would like to utilize the entire dataset for learning the predictor; we leave that for future work.

While most of the techniques we use are standard [3], the Lipschitz continuity of the robust loss function requires a more delicate analysis. For that, we have the following result.

**Lemma 4.2** (Parameter adversarial stability). *Let* $\mathbf{D}, \mathbf{D}' \in \mathcal{D}$. *If* $\|\mathbf{D} - \mathbf{D}'\|_2 \leq \epsilon \leq 2\lambda/(1+\nu)^2$, *then*

$$\max_{\mathbf{v} \in \Delta} \|\varphi_{\mathbf{D}}(\mathbf{x} + \mathbf{v}) - \varphi_{\mathbf{D}'}(\mathbf{x} + \mathbf{v})\|_2 \leq \gamma(1+\nu)^2 \epsilon, \tag{2}$$

*with* $\gamma = \frac{3}{2} \frac{\sqrt{s}}{\lambda(1-\eta_s)}$, *as long as* $\tau_s(\mathbf{x}) \geq 2\nu + \sqrt{\epsilon} \left( \sqrt{\frac{25}{\lambda}}(1+\nu) + 2\left(\frac{(1+\nu)}{\lambda} + 1\right) \right)$.

Lemma 4.2 is central to our proof, as it provides a bound on difference between the features computed by the encoder under model deviations. Note that the condition on the minimal encoder gap, $\tau_s(\mathbf{x})$, puts an upper bound on the distance between models $\mathbf{D}$ and $\mathbf{D}'$. This in turn results in the condition imposed on the minimal samples in Theorem 4.1. It is worth stressing that the lower bound on $\tau_s(\mathbf{x})$ is on the *unperturbed* encoder gap – that which can be evaluated on the samples from the dataset, without the need of the adversary. We defer the proof of this Lemma to Appendix B.1.

## 5  Robustness Certificate

Next, we turn to address an equally important question about robust adversarial learning, that of giving a formal certification of robustness. Formally, we would like to guarantee that the output of the trained model, $f_{\mathbf{D},\mathbf{w}}(\mathbf{x})$, does not change for norm-bounded adversarial perturbations of a certain size. Our second main result provides such a certificate for the supervised sparse coding model.

Here, we consider a multiclass classification setting with $y \in \{1, \ldots, K\}$; simplified results for binary classification are included in Appendix C. The hypothesis class is parameterized as $f_{\mathbf{D},\mathbf{W}}(\mathbf{x}) = \mathbf{W}^T \varphi_{\mathbf{D}}(\mathbf{x})$, with $\mathbf{W} = [\mathbf{W}_1, \mathbf{W}_2, \ldots, \mathbf{W}_K] \in \mathbb{R}^{p \times K}$. The multiclass margin is defined as follows:

$$\rho_{\mathbf{x}} = \mathbf{W}_{y_i}^T \varphi_{\mathbf{D}}(\mathbf{x}) - \max_{j \neq y_i} \mathbf{W}_j^T \varphi_{\mathbf{D}}(\mathbf{x}).$$

We show the following result.

**Theorem 5.1** (Robustness certificate for multiclass supervised sparse coding). *Let* $\rho_x > 0$ *be the multiclass classifier margin of* $f_{\mathbf{D},\mathbf{w}}(\mathbf{x})$ *composed of an encoder with a gap of* $\tau_s(\mathbf{x})$ *and a dictionary,* $\mathbf{D}$, *with RIP constant* $\eta_s < 1$. *Let* $c_{\mathbf{W}} := \max_{i \neq j} \|\mathbf{W}_i - \mathbf{W}_j\|_2$. *Then,*

$$\arg\max_{j \in [K]} [\mathbf{W}^T \varphi_{\mathbf{D}}(\mathbf{x})]_j = \arg\max_{j \in [K]} [\mathbf{W}^T \varphi_{\mathbf{D}}(\mathbf{x} + \mathbf{v})]_j, \quad \forall \mathbf{v} : \|\mathbf{v}\|_2 \leq \nu, \tag{3}$$

*so long as* $\nu \leq \min\{\tau_s(\mathbf{x})/2, \rho_{\mathbf{x}}\sqrt{1 - \eta_s}/c_{\mathbf{W}}\}$.

Theorem 5.1 clearly captures the potential contamination on two flanks: robustness can no longer be guaranteed as soon as the energy of the perturbation is enough to either significantly modify the computed representation *or* to induce a perturbation larger than the classifier margin on the feature space. Proof of Theorem 5.1, detailed in Appendix C, relies on the following lemma showing that under an encoder gap assumption, the computed features are moderately affected despite adversarial corruptions of the input vector.

**Lemma 5.2** (Stability of representations under adversarial perturbations). *Let* $\mathbf{D}$ *be a dictionary with RIP constant* $\eta_s$. *Then, for any* $\mathbf{x} \in \mathcal{X}$ *and its additive perturbation* $\mathbf{x} + \mathbf{v}$, *for any* $\|\mathbf{v}\|_2 \leq \nu$, *if* $\tau_s(\mathbf{x}) > 2\nu$, *then we have that*

$$\|\varphi_{\mathbf{D}}(\mathbf{x}) - \varphi_{\mathbf{D}}(\mathbf{x} + \mathbf{v})\|_2 \leq \frac{\nu}{\sqrt{1 - \eta_s}}. \tag{4}$$

An extensive set of results exist for the stability of the solution provided by Lasso relying generative model assumptions [Foucart and Rauhut, 2017, Elad, 2010]. The novelty of Lemma 5.2 is in replacing such an assumption with the existence of a positive encoder gap on $\varphi_{\mathbf{D}}(\mathbf{x})$.

Going back to Theorem 5.1, note that the upper bound on $\nu$ depends on the RIP constant $\eta_s$, which is not computable for a given (deterministic) matrix $\mathbf{D}$. Yet, this result can be naturally relaxed by upper bounding $\eta_s$ with measures of correlation between the atoms, such as the mutual coherence. This quantity provides a measure of the worst correlation between two atoms in the dictionary $\mathbf{D}$, and it is defined as $\mu(\mathbf{D}) = \max_{i \neq j} |\langle \mathbf{D}_i, \mathbf{D}_j \rangle|$ (for $\mathbf{D}$ with normalized columns). For general (overcomplete and full rank) dictionaries, clearly $0 < \mu(\mathbf{D}) \leq 1$.

While conceptually simple, results that use $\mu(\mathbf{D})$ tend to be too conservative. Tighter bounds on $\eta_s$ can be provided by the Babel function[4], $\mu_{(s)}$, which quantifies the maximum correlation between an atom and *any other* collection of $s$ atoms in $\mathbf{D}$. It can be shown [Tropp et al., 2003, Elad, 2010, Chapter 2] that $\eta_s \leq \mu_{(s-1)} \leq (s-1)\mu(\mathbf{D})$. Therefore, we have the following:

**Corollary 5.3.** *Under the same assumptions as those in Theorem 5.1,*

$$\arg\max_{j\in[K]} \ [\mathbf{W}^T\varphi_{\mathbf{D}}(\mathbf{x})]_j \ = \arg\max_{j\in[K]} \ [\mathbf{W}^T\varphi_{\mathbf{D}}(\mathbf{x}+\mathbf{v})]_j, \quad \forall\, \mathbf{v} : \|\mathbf{v}\|_2 \leq \nu \qquad (5)$$

*so long as* $\nu \leq \min\{\tau_s(\mathbf{x})/2, \rho_{\mathbf{x}}\sqrt{1-\mu_{(s-1)}}/c_{\mathbf{W}}\}$.

Although the condition on $\nu$ in the corollary above is stricter (and the bound looser), the quantities involved can easily be computed numerically leading to practical useful bounds, as we see next.

## 6  Experiments

In this section, we illustrate the robustness certificate guarantees both in synthetic and real data, as well as the trade-offs between constants in our sample complexity result. First, we construct samples from a separable binary distribution of $k$-sparse signals. To this end, we employ a dictionary with 120 atoms in 100 dimensions with a mutual coherence of 0.054. Sparse representations $\mathbf{z}$ are constructed by first drawing their support (with cardinality $k$) uniformly at random, and drawing its non-zero entries from a uniform distribution away from zero. Samples are obtained as $\mathbf{x} = \mathbf{D}\mathbf{z}$, and normalized to unit norm. We finally enforce separability by drawing $\mathbf{w}$ at random from the unit ball, determining the labels as $y = \text{sign}(\mathbf{w}^T\varphi_{\mathbf{D}}(\mathbf{x}))$, and discarding samples with a margin $\rho$ smaller than a pre-specified amount (0.05 in this case). Because of the separable construction, the accuracy of the resulting classifier is 1.

We then attack the obtained model employing the projected gradient descent method [Madry et al., 2017], and analyze the degradation in accuracy as a function of the energy budget $\nu$. We compare this empirical performance with the bound in Corollary 5.3: given the obtained margin, $\rho$, and the dictionary's $\mu_s$, we can compute the maximal certified radius for a sample $\mathbf{x}$ as

$$\nu(\mathbf{x}) = \max_s \min\{\tau_s(\mathbf{x})/2, \rho_{\mathbf{x}}\sqrt{1-\mu_{(s-1)}}/c_{\mathbf{W}}\}. \qquad (6)$$

For a given dataset, we can compute the minimal certified radius over the samples, $\nu^* = \min_{i\in[n]} \nu(\mathbf{x}_i)$. This is the bound depicted in the vertical line in Figure 2a. As can be seen, despite being somewhat loose, the attacks do not change the label of the samples, thus preserving the accuracy.

In non-separable distributions, one may study how the accuracy depends on the *soft margin* of the classifier. In this way, one can determine a target margin that results in, say, 75% accuracy on a validation set. One can obtain a corresponding certified radius of $\nu^*$ as before, which will guarantee that the accuracy will not drop below 75% as long as $\nu < \nu^*$. This is illustrated in Figure 2b.

An alternative way of employing our results from Section 5 is by studying the *certified accuracy* achieved by the resulting hypothesis. The certified accuracy quantifies the percentage of samples in a test set that are classified correctly while being *certifiable*. In our context, this implies that a sample $\mathbf{x}$ achieves a margin of $\rho_{\mathbf{x}}$, for which a certified radius of $\nu^*$ can be obtained with (6). In this way, one may study how the certified accuracy decreases with increasing $\nu^*$.

This analysis lets us compare our bounds with those of other popular certification techniques, such as randomized smoothing [Cohen et al., 2019]. Randomized smoothing provides high probability robustness guarantees against $\ell_2$ attacks for *any* classifier by composing them with a Gaussian distribution (though other distributions have been recently explored as well for other $l_p$ norms [Salman et al., 2020]). In a nutshell, the larger the variance of the Gaussian, the larger the certifiable radius becomes, albeit at the expense of a drop in accuracy.

We use the MNIST dataset for this analysis. We train a model with 256 atoms by minimizing the following regularized empirical risk using stochastic gradient descent (employing Adam [Kingma

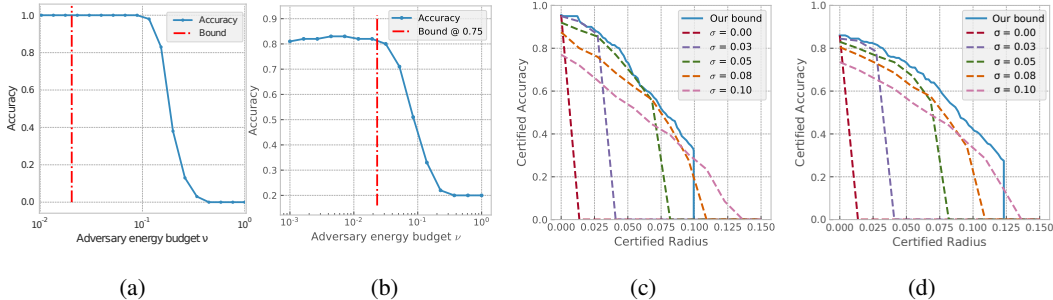

Figure 2: Numerical demonstrations of our results. (a) synthetic separable distribution. (b) synthetic non-separable distribution. (c-d) certified accuracy on MNIST with $\lambda = 0.2$ and $\lambda = 0.3$, respectively, comparing with Randomized Smoothing with different variance levels.

and Ba, 2014]; the implementation details are deferred to Appendix D)

$$\min_{\mathbf{W}, \mathbf{D}} \ \frac{1}{m} \sum_{i=1}^{m} \ell\left(y_i, \langle \mathbf{W}, \varphi_{\mathbf{D}}(\mathbf{x}_i) \rangle\right) + \alpha \|\mathbf{I} - \mathbf{D}^T \mathbf{D}\|_F^2 + \beta \|\mathbf{W}\|_F^2, \tag{7}$$

where $\ell$ is the cross entropy loss. Recall that $\varphi_{\mathbf{D}}(\mathbf{x})$ depends on $\lambda$, and we train two different models with two values for this parameter ($\lambda = 0.2$ and $\lambda = 0.3$).

Figure 2c and 2d illustrate the certified accuracy on 200 test samples obtained by different degrees of randomized smoothing and by our result. While the certified accuracy resulting from our bound is comparable to that by randomized smoothing, the latter provides a certificate by *defending* (i.e. composing it with a Gaussian distribution). In other words, different *smoothed* models have to be constructed to provide different levels of certified accuracy. In contrast, our model is not defended or modified in any way, and the certificate relies solely on our careful characterization of the function class. Since randomized smoothing makes no assumptions about the model, the bounds provided by this strategy rely on the estimation of the output probabilities. This results in a heavy computational burden to provide a high-probability result (a failure probability of 0.01% was used for these experiments). In contrast, our bound is deterministic and trivial to compute.

Lastly, comparing the results in Figure 2c (where $\lambda = 0.2$) and Figure 2d (where $\lambda = 0.3$), we see the trade-off that we alluded to in Section 4: larger values of $\lambda$ allow for larger encoder gaps, resulting in overall larger possible certified radius. In fact, $\lambda$ determines a hard bound on the possible achieved certified radius, given by $\lambda/2$, as per (6). This, however, comes at the expense of reducing the complexity of the representations computed by the encoder $\varphi_{\mathbf{D}}(\mathbf{x})$, which impacts the risk attained.

## 7 Conclusion

In this paper we study the adversarial robustness of the supervised sparse coding model from two main perspectives: we provide a bound for the robust risk of any hypothesis that achieves a minimum encoder gap over the samples, as well as a robustness certificate for the resulting end-to-end classifier. Our results describe guarantees relying on the interplay between the computed representations, or features, and the classifier margin.

While the model studied is still relatively simple, we envision several ways in which our analysis can be extended to more complex models. First, high dimensional data with shift-invariant properties (such as images) often benefit from convolutional features. Our results do hold for convolutional dictionaries, but the conditions on the mutual coherence may become prohibitive in this setting. An analogous definition of encoder gap in terms of convolutional sparsity [Papyan et al., 2017b] may provide a solution to this limitation. Furthermore, this analysis could also be extended to sparse models with multiple layers, as in [Papyan et al., 2017a, Sulam et al., 2019]. On the other hand, while our result does not provide a uniform learning bound over the hypothesis class, we have found empirically that regularized ERM does indeed return hypotheses satisfying non-trivial encoder gaps. The theoretical underpinning of this phenomenon needs further research. More generally, even though this work focuses on sparse encoders, we believe similar principles could be generalized to other forms of representations in a supervised learning setting, providing a framework for the principled analysis of adversarial robustness of machine learning models.

## Broader Impact

This work contributes to the theoretical understanding of the limitations and achievable robustness guarantees for supervised learning models. Our results can therefore provide tools that could be deployed in sensitive settings where these types of guarantees are a priority. On a broader note, this work advocates for the precise analysis and characterization of the data-driven features computed by modern machine learning models, and we hope our results facilitate their generalization to other more complex models.

## Acknowledgements

This research was supported, in part, by DARPA GARD award HR00112020004, NSF BIGDATA award IIS-1546482, NSF CAREER award IIS-1943251 and NSF TRIPODS award CCF-1934979. Jeremias Sulam kindly thanks Aviad Aberdam for motivating and inspiring discussions. Raman Arora acknowledges support from the Simons Institute as part of the program on the Foundations of Deep Learning and the Institute for Advanced Study (IAS), Princeton, NJ, as part of the special year on Optimization, Statistics, and Theoretical Machine Learning.

## Footnotes

[1] The solution is unique under mild assumptions [Tibshirani et al., 2013], and otherwise our results hold for any solution returned by a deterministic solver.

[2] Code to reproduce all of our experiments is made available at our github repository.

[3]See [Seibert, 2019] for a comprehensive review on these tools in matrix factorization problems.

[4]Let $\Lambda$ denote subsets (supports) of $\{1, 2, \ldots, p\}$. Then, the Babel function is defined as $\mu_{(s)} = \max_{\Lambda:|\Lambda|=s} \max_{j\notin\Lambda} \sum_{i\in\Lambda} |\langle\mathbf{D}_i, \mathbf{D}_j\rangle|$.

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
