[Supplementary Material]

# Supplementary Material for
# Adversarial Robustness of Supervised Sparse Coding

## A  Encoder Gap for $k$-sparse signals

Herein we show that a positive encoder gap exists for signals that are (approximately) $k$-sparse. Consider signals $\mathbf{x}$ obtained as $\mathbf{x} = \mathbf{D}\mathbf{z} + \mathbf{v}$, where $\mathbf{D} \in \mathcal{D}$, $\|\mathbf{v}\|_2 \leq \nu$ and $\mathbf{z}$ is sampled from a distribution of sparse vectors with up to $k$ non-zeros, with $k < \frac{1}{3}\left(1 + \frac{1}{\mu(\mathbf{D})}\right)$, where $\mu(\mathbf{D}) = \max_{i \neq j}\langle \mathbf{D}_i, \mathbf{D}_j \rangle$ is the mutual coherence of $\mathbf{D}$. Then, from [Tropp, 2006], if $\lambda = 4\nu$, the (unique) solution recovered by $\boldsymbol{\alpha} = \varphi_{\mathbf{D}}(\mathbf{x} + \mathbf{v})$ satisfies $\|\boldsymbol{\alpha} - \mathbf{z}\|_\infty \leq \frac{15}{2}\nu$, and $Supp(\boldsymbol{\alpha}) \subseteq Supp(\mathbf{z})$. Recall the definition of encoder gap:

$$\tau_s(\mathbf{x}) \coloneqq \max_{\mathcal{I} \in \Lambda^{p-s}} \min_{i \in \mathcal{I}} \ (\lambda - |\langle \mathbf{D}_i, \mathbf{x} - \mathbf{D}\varphi_{\mathbf{D}}(\mathbf{x})\rangle|)$$

and pick $s > k$. Let $S = Supp(\mathbf{z})$. Thus, the maximization over $I$ is achieved by a subset $\mathcal{I}$ which does not contain any of the active atoms in $\mathbf{z}$ (for which $|\langle \mathbf{D}_i, \mathbf{x} - \mathbf{D}\varphi_{\mathbf{D}}(\mathbf{x})\rangle| = \lambda$, by optimality).

Now, define $\Delta = \boldsymbol{\alpha} - \mathbf{z}$, and let $\Delta_S$ denote the vector $\Delta$ restricted to the support $S$ and $\mathbf{D}_S$ the sub-dictionary obtained $\mathbf{D}$ by restricting it to the same set of atoms. We can then write

$$\max_i \left|\mathbf{D}_i^T(\mathbf{x} - \mathbf{D}\boldsymbol{\alpha})\right| = \max_i \left|\mathbf{D}_i^T(\mathbf{x} - \mathbf{D}\mathbf{z}) - \mathbf{D}_i^T\mathbf{D}\Delta\right| \tag{8}$$

$$= \max_i \left|\mathbf{D}_i^T\mathbf{D}_S\Delta_S\right| \tag{9}$$

$$\leq \max_i |\mathbf{D}_i^T\mathbf{D}_S|\|\Delta_S\|_\infty \tag{10}$$

$$= \frac{15}{2}\mu(\mathbf{D})\nu, \tag{11}$$

because $i \notin S$. Thus,

$$\min_i \lambda - |\langle \mathbf{D}_i, \mathbf{x} - \mathbf{D}\varphi_{\mathbf{D}}(\mathbf{x})\rangle| \geq \lambda - \frac{15}{2}\mu(\mathbf{D})\nu. \tag{12}$$

In fact, recalling that $\lambda = 4\nu$, we have that $\tau_s \geq \nu(4 - \mu(\mathbf{D})\frac{15}{2})$.

## B  Robust Generalization Bound

Herein we prove our generalization bound, but first re-state it for completeness.

**Theorem 4.1.** *Let $\mathcal{W} = \{\mathbf{w} \in \mathbb{R}^p : \|\mathbf{w}\|_2 \leq B\}$, and $\mathcal{D}$ be the set of column-normalized dictionaries with $p$ columns and with RIP at most $\eta_s^*$. Let $\mathcal{H} = \{f_{\mathbf{D},\mathbf{w}}(\mathbf{x}) = \langle \mathbf{w}, \varphi_{\mathbf{D}}(\mathbf{x})\rangle : \mathbf{w} \in \mathcal{W}, \mathbf{D} \in \mathcal{D}\}$. Denote $\tau_s^*$ the minimal encoder gap over the $m$ samples. Then, with probability at least $1 - \delta$ over the draw of the $m$ samples, the generalization gap for any hypothesis $f \in \mathcal{H}$ that achieves an encoder gap of $\tau_s^* > 2\nu$, satisfies*

$$\left|\tilde{R}_S(f) - \tilde{R}(f)\right| \leq \frac{b}{\sqrt{m}}\left((d+1)p\log\left(\frac{3m}{2\lambda(1 - \eta_s^*)}\right) + p\log(B) + \log\frac{4}{\delta}\right)^{\frac{1}{2}}$$

$$+ b\sqrt{\frac{2\log(m/2) + 2\log(2/\delta)}{m}} + 12\frac{(1 + \nu)^2 L_\ell B\sqrt{s}}{m},$$

*as long as $m > \frac{\lambda(1-\eta_s)}{(\tau_s^* - 2\nu)^2}K_\lambda$, where $K_\lambda = \left(2\left(1 + \frac{1+\nu}{2\lambda}\right) + \frac{5(1+\nu)}{\sqrt{\lambda}}\right)^2$.*

*Proof.* Fix $\epsilon > 0$, and consider a minimal $\epsilon$-cover for the parameter space $(\mathcal{D}, \mathcal{W})$ with respect to a metric $d$ and with the elements $(\mathbf{D}_j, \mathbf{w}_j)$, $j \in \{1, \ldots, N^{cov}((\mathcal{D}, \mathcal{W}), \epsilon)\}$. The metric we consider is the max over the operator norm and $\ell_2$ norm on $\mathcal{D}$ and $\mathcal{W}$, respectively, i.e. $d((\mathbf{D}, \mathbf{w}), (\mathbf{D}', \mathbf{w}')) = \max\{\|\mathbf{D} - \mathbf{D}'\|_2, \|\mathbf{w} - \mathbf{w}'\|_2\}$. Now, fixing $(\mathbf{D}, \mathbf{w})$, by the definition of the $\epsilon$-cover, there exist an

index $j$ so that $d((\mathbf{D}_j, \mathbf{w}_j), (\mathbf{D}, \mathbf{w})) \le \epsilon$. We thus expand the generalization gap into three terms, as follows:

$$\left| \tilde{R}_S(f) - \tilde{R}(f) \right| = \left| \frac{1}{m} \sum_{i=1}^{m} \tilde{\ell}_\nu(y_i, f(\mathbf{x}_i)) - \mathop{\mathbb{E}}_{(\mathbf{x},y) \sim P} \left[ \tilde{\ell}_\nu(y, f(\mathbf{x})) \right] \right| \tag{13}$$

$$\le \sup_{k \in [N^{\mathrm{cov}}]} \left| \frac{1}{m} \sum_{i=1}^{m} \tilde{\ell}_\nu(y_i, f_{\mathbf{D}_k, \mathbf{w}_k}(\mathbf{x}_i)) - \mathop{\mathbb{E}}_{(\mathbf{x},y)} \left[ \tilde{\ell}_\nu(y, f_{\mathbf{D}_k, \mathbf{w}_k}(\mathbf{x})) \right] \right| \tag{14}$$

$$+ \left| \frac{1}{m} \sum_{i=1}^{m} \tilde{\ell}_\nu(y_i, f_{\mathbf{D}, \mathbf{w}}(\mathbf{x}_i)) - \frac{1}{m} \sum_{i=1}^{m} \tilde{\ell}_\nu(y_i, f_{\mathbf{D}_j, \mathbf{w}_j}(\mathbf{x}_i)) \right| \tag{15}$$

$$+ \left| \mathop{\mathbb{E}}_{(\mathbf{x},y)} \left[ \tilde{\ell}_\nu(y, f_{\mathbf{D}_j, \mathbf{w}_j}(\mathbf{x})) \right] - \mathop{\mathbb{E}}_{(\mathbf{x},y)} \left[ \tilde{\ell}_\nu(y, f_{\mathbf{D}, \mathbf{w}}(\mathbf{x})) \right] \right|. \tag{16}$$

Let us bound the first of these terms. Let $\mathbf{z}_i$ denote the random tuple $(y_i, \mathbf{x}_i)$, and $\mathbf{Z} = [(y_1, \mathbf{x}_1), \ldots, (y_m, \mathbf{x}_m)]$. Let $g(\mathbf{Z}) = \frac{1}{m} \sum_{i=1}^{m} \tilde{\ell}_f(y_i, \mathbf{x}_i)$. Furthermore, consider $\mathbf{Z}'$ as the set of $m$ random variables $\mathbf{z}$ that only differs from $\mathbf{Z}$ in its $i^{th}$ variable, $\mathbf{z}'_i = (y'_i, \mathbf{x}'_i)$. Then, for any $i \in [m]$,

$$|g(\mathbf{Z}) - g(\mathbf{Z}')| = \left| \frac{1}{m} \left( \tilde{\ell}_{\nu_f}(y_i, \mathbf{x}_i) - \tilde{\ell}_{\nu_f}(y'_i, \mathbf{x}'_i) \right) \right| \le \frac{b}{m}, \tag{17}$$

since $\ell(y, f(\mathbf{x}))$, and thus $\tilde{\ell}(y, f(\mathbf{x}))$, is bounded.

$$\Pr \left[ \, |g(\mathbf{Z}) - \mathbb{E}[g(\mathbf{Z})]| \ge t \, \right] \le 2 \exp \left( \frac{-2mt^2}{b^2} \right). \tag{18}$$

Furthermore, note that $\mathbb{E}[g(\mathbf{Z})] = \mathbb{E}[\tilde{\ell}_{\nu_f}(y, \mathbf{x})]$ (linearity of expectation), and thus we have that

$$\Pr \left[ \left| \frac{1}{m} \sum_{i=1}^{m} \tilde{\ell}_\nu(y_i, f_{\mathbf{D}, \mathbf{w}}(\mathbf{x}_i)) - \mathop{\mathbb{E}}_{(\mathbf{x},y)} \tilde{\ell}_\nu(y, f_{\mathbf{D}, \mathbf{w}}(\mathbf{x})) \right| > t \right] \le 2 \exp \left( \frac{-2mt^2}{b^2} \right). \tag{19}$$

Next, using a union bound argument, we can bound the probability over the supremum:

$$\Pr \left[ \sup_j \left| \frac{1}{m} \sum_{i=1}^{m} \tilde{\ell}(y_i, f_{\mathbf{D}_j, \mathbf{w}_j}(\mathbf{x}_i)) - \mathop{\mathbb{E}}_{(\mathbf{x},y)} \tilde{\ell}(y, f_{\mathbf{D}_j, \mathbf{w}_j}(\mathbf{x})) \right| > t \right] \le$$

$$\sum_{j=1}^{N^{cov}} \Pr \left[ \left| \frac{1}{m} \sum_{i=1}^{m} \tilde{\ell}(y_i, f_{\mathbf{D}_j, \mathbf{w}_j}(\mathbf{x}_i)) - \mathop{\mathbb{E}}_{(\mathbf{x},y)} \tilde{\ell}(y, f_{\mathbf{D}_j, \mathbf{w}_j}(\mathbf{x})) \right| > t \right] \le 2 N^{cov} \exp \left( \frac{-2mt^2}{b^2} \right). \tag{20}$$

Denote this failure probability as $\delta/2$. Thus, with probability at least $1 - \delta/2$, we get

$$\sup_j \left| \frac{1}{m} \sum_{i=1}^{m} \tilde{\ell}(y_i, f_{\mathbf{D}_j, \mathbf{w}_j}(\mathbf{x}_i)) - \mathop{\mathbb{E}}_{(\mathbf{x},y)} \left[ \tilde{\ell}(y, f_{\mathbf{D}_j, \mathbf{w}_j}(\mathbf{x})) \right] \right| \le b \sqrt{\frac{\log(N^{cov}) + \log(4/\delta)}{2m}}. \tag{21}$$

Let us now focus on the second and third terms in Eq. (14). In particular, we will upper bound them by analyzing the Lipschitz continuity of the loss function with respect to the parameters, $\mathbf{D}$ and $\mathbf{w}$. We assume that $\ell$ is $L_\ell$-Lipschitz, and we analyze the Lipschitz continuity of $\tilde{\ell}$ w.r.t $\mathbf{D}$ through $f_\mathbf{D}(\mathbf{x})$. Noting that the difference of the maxima is upper-bounded by the maximum of the difference, we can write

$$\left| \tilde{\ell}(y, f_\mathbf{D}(\mathbf{x})) - \tilde{\ell}(y, f_{\mathbf{D}'}(\mathbf{x})) \right| = \left| \max_{\mathbf{v} \in \Delta} \ell(y, f_\mathbf{D}(\mathbf{x} + \mathbf{v})) - \max_{\mathbf{v} \in \Delta} \ell(y, f_{\mathbf{D}'}(\mathbf{x} + \mathbf{v})) \right| \tag{22}$$

$$\le \max_{\mathbf{v} \in \Delta} \left| \ell(y, f_\mathbf{D}(\mathbf{x} + \mathbf{v})) - \ell(y, f_{\mathbf{D}'}(\mathbf{x} + \mathbf{v})) \right| \tag{23}$$

$$\le L_\ell \max_{\mathbf{v} \in \Delta} \left| \langle \mathbf{w}^T, \varphi_\mathbf{D}(\mathbf{x} + \mathbf{v}) \rangle - \langle \mathbf{w}^T, \varphi_{\mathbf{D}'}(\mathbf{x} + \mathbf{v}) \rangle \right| \tag{24}$$

$$\le L_\ell \|\mathbf{w}\|_2 \max_{\mathbf{v} \in \Delta} \|\varphi_\mathbf{D}(\mathbf{x} + \mathbf{v}) - \varphi_{\mathbf{D}'}(\mathbf{x} + \mathbf{v})\|_2. \tag{25}$$

We will now bound the term $\max_{\mathbf{v} \in \Delta} \|\varphi_{\mathbf{D}}(\mathbf{x} + \mathbf{v}) - \varphi_{\mathbf{D}'}(\mathbf{x} + \mathbf{v})\|_2$. Notice that if the dictionary $\mathbf{D}$ has an encoder gap of at least $\tau_s^*$ for an input sample $\mathbf{x}$, then we can use Lemma 4.2 to obtain

$$\max_{\mathbf{v} \in \Delta} \|\varphi_{\mathbf{D}}(\mathbf{x} + \mathbf{v}) - \varphi_{\mathbf{D}'}(\mathbf{x} + \mathbf{v})\|_2 \leq \frac{3}{2} \frac{\sqrt{s}(1+\nu)^2}{\lambda(1-\eta_s)} \epsilon.$$

Denote the probability of this event (that $\tau_s(\mathbf{x}) > \tau_s^*$) by $1 - \rho$. Note that $\epsilon \leq 2\lambda/(1+\nu)^2$ is required in order to apply Lemma 4.2, but this condition is mild and we will later show that this holds under the condition of minimal number of samples.

Likewise, $\tilde{\ell}$ is Lipschitz continuous w.r.t $\mathbf{w}$,

$$\left|\tilde{\ell}(y, f_{\mathbf{D}, \mathbf{w}}(\mathbf{x})) - \tilde{\ell}(y, f_{\mathbf{D}, \mathbf{w}'}(\mathbf{x}))\right| \leq L_\ell \max_{\mathbf{v} \in \Delta} |\langle \mathbf{w}^T, \varphi_{\mathbf{D}}(\mathbf{x} + \mathbf{v}) \rangle - \langle \mathbf{w}'^T, \varphi_{\mathbf{D}}(\mathbf{x} + \mathbf{v}) \rangle| \quad (26)$$

$$\leq \frac{L_\ell(1+\nu)^2}{\lambda} \|\mathbf{w} - \mathbf{w}'\|_2, \quad (27)$$

since $\max_{\mathbf{v} \in \Delta} \|\varphi_{\mathbf{D}}(\mathbf{x} + \mathbf{v})\|_2 = (1+\nu)^2/\lambda$ ( Remark B.2). Furthermore, $\|\mathbf{w} - \mathbf{w}'\|_2 \leq \epsilon$, as follows from our definition of $\epsilon$-cover.

On the other hand, if $\mathbf{D}$ does not achieve this encoder gap ($\tau_s(\mathbf{x}) < \tau^*$), which happens with probability $\rho$, then we can simply upper bound the worst possible loss, i.e. $\left|\tilde{\ell}(y, f_{\mathbf{D}}(\mathbf{x})) - \tilde{\ell}(y, f_{\mathbf{D}'}(\mathbf{x}))\right| \leq b$.

Let us now analyze this probability, $\rho$. For simplicity, assume that $\tau_s(\mathbf{x}_i)$ are i.i.d. random variables. e.g. by computing $\tau_s(\mathbf{x}_i)$ on a held-out set with $m_2$ samples, independent from the $m_1$ samples that are used to train the dictionary. In particular, we split training and development samples $m_1$ and $m_2$ equally $m_1 = m_2 = m/2$. Let $F_{m_2}(\tau) := \frac{1}{m_2} \sum_{i=1}^{m_2} 1_{\{\tau_s(\mathbf{x}_i) < \tau\}}$ denote the fraction of training points that achieve the encoder margin smaller than $\tau$. Let $F(\tau) := \Pr(\tau_s(\mathbf{x}) < \tau)$. Then, uniform convergence [Mohri et al., 2018] yields that for any $\delta/2 > 0$, with probability at least $1 - \delta/2$, we have that $\sup_{\tau \in \mathbb{R}} |F_{m_2}(\tau) - F(\tau)| \leq c\sqrt{\frac{\log(m_2) + \log(2/\delta)}{m_2}}$, for some constant $c$. Since this holds uniformly for any $\tau$, it holds in particular for $\tau = \tau_s^*$. This implies then that $F(\tau) = \Pr(\tau_s(\mathbf{x}) \leq \tau^*) \leq c\sqrt{\frac{\log(m_2) + \log(1/\delta_2)}{m_2}} = \rho$.

Note that the third term in Eq. (14) involves the expectation over the population, and so we can upper bound that term by

$$\frac{L_\ell(1+\nu)^2}{\lambda}\left(1 + \frac{3}{2}\frac{B\sqrt{s}}{(1-\eta_s)}\right)\epsilon + \frac{b}{\sqrt{m_2}}c\left(\sqrt{\log(m_2) + \log(2/\delta)}\right),$$

with probability at least $1 - \delta/2$. The second term, on the other hand, is the average loss over the training samples. For this, it suffices to note that the uniform bound $\sup_{\tau \in \mathbb{R}} |F_{m_2}(\tau) - F(\tau)|$ holds for *any* choice of $\tau$. In particular, it holds for $\tau_s^*$ defined over both training and development samples. As a result, the dictionary satisfies the encoder gap on those samples, and so the second term can be simply upper bounded by

$$\frac{L_\ell(1+\nu)^2}{\lambda}\left(1 + \frac{3}{2}\frac{B\sqrt{s}}{(1-\eta_s)}\right)\epsilon.$$

We finally get expressions for the covering number as a function of $\epsilon$. For oblique manifolds (matrices of size $n \times p$ with unit norm columns), $N^{cov}(\mathcal{D}, \epsilon) \leq (3/\epsilon)^{dp}$, while for $B$-bounded vectors $N^{cov}(\mathcal{W}, \epsilon) \leq (3B/\epsilon)^p$ [Seibert, 2019]. Thus, the covering number of the direct product of the two constraint sets can be bounded by $N^{cov}(\mathcal{D}, \mathcal{W}, \epsilon) \leq (3/\epsilon)^{(d+1)p} B^p$.

Gathering everything together, we can bound the generalization error by

$$\left|\tilde{R}_S(f) - \tilde{R}(f)\right| \leq b\sqrt{\frac{(d+1)p\log(3/\epsilon) + p\log(B) + \log(4/\delta)}{m}} + bc\sqrt{2\frac{\log(m/2) + \log(2/\delta)}{m}}$$

$$+ \frac{2L_\ell(1+\nu)^2}{\lambda}\left(1 + \frac{3}{2}\frac{B\sqrt{s}}{(1-\eta_s)}\right)\epsilon. \quad (28)$$

All that remains is to set $\epsilon$ appropriately. Set $\epsilon = \lambda(1 - \eta_s)/m$, and so

$$\left| \tilde{R}_S(f) - \tilde{R}(f) \right| \leq b\sqrt{\frac{(d+1)p\log(3m/(2\lambda(1 - \eta_s))) + p\log(B) + \log 4/\delta}{m}}$$
$$+ bc\sqrt{2\frac{\log(m/2) + \log(2/\delta)}{m}} + 12\frac{B\sqrt{s}L_\ell(1 + \nu)^2}{m}. \quad (29)$$

Lastly, the results above holds for $\epsilon$ small enough. Due to Lemma Lemma B.6, one needs

$$\tau_s^* > 2\nu + \sqrt{\epsilon}\left(\sqrt{\frac{25}{\lambda}}(1 + \nu) + 2\left(\frac{(1 + \nu)}{2\lambda} + 1\right)\right).$$

implying that

$$\sqrt{\frac{\lambda(1 - \eta_s)}{m}} < \frac{\tau_s - 2\nu}{2\left(1 + \frac{1+\nu}{2\lambda}\right) + (1 + \nu)\sqrt{\frac{25}{\lambda}}}.$$

Recalling that, naturally, $0 < \sqrt{1/m}$, it is enough to require that $\tau_s > 2\nu$ and that

$$m > \frac{\lambda(1 - \eta_s)}{(\tau_s - 2\nu)^2}\left(2\left(1 + \frac{1 + \nu}{2\lambda}\right) + (1 + \nu)\sqrt{\frac{25}{\lambda}}\right)^2. \quad (30)$$

Lastly, we need to show that this condition guarantees the assumption $\epsilon \leq 2\lambda/(1 + \nu)^2$ is satisfied, in order to apply Lemma 4.2 above. Note that $\epsilon = \lambda(1 - \eta_s)/m \leq \lambda/m$. Thus, we need $\lambda/m \leq 2\lambda/(1 + \nu)^2$, which is satisfied as long as $m \geq 2 \geq (1 + \nu)^2/2$, which is satisfied in all relevant scenarios.

$\square$

## B.1  Parameter adversarial stability

In this section we prove the key result in Lemma 4.2, guaranteeing that the perturbation in the encoded features under model deviations and adversarial contamination is bounded. The main difficulty here is that the Lasso encoder solves a problem that is not strongly convex – due to the overcompleteness of the dictionary – and thus showing that the encoded features satisfy a Lipschitz property w.r.t the model parameters, particularly in the adversarial setting, is not trivial. As a result, this section will be dedicated to showing that if the model perturbation and adversarial contamination is small enough and there exist a positive encoder margin, then some sparsity is retained in the features after the respective perturbations. With this result at hand, the proof of Lemma 4.2 follows directly from the proof of Theorem 4 in [Mehta and Gray, 2013], albeit with different constants which account for the perturbation $\mathbf{v}$. This *preservation of sparsity* result is formalized later in Lemma B.6, and the following immediate lemmata will build some intermediate results needed for it.

We first make a few remarks about the encoded features. We assume that $\mathbf{x} \in \mathcal{X} = \{\mathbf{x} : \|\mathbf{x}\|_2 \leq 1\}$, and recall that $\varphi_{\mathbf{D}}(\mathbf{x}) = \arg\min_{\mathbf{z}} \frac{1}{2}\|\mathbf{x} - \mathbf{D}\mathbf{z}\|_2^2 + \lambda\|\mathbf{z}\|_1$. We are interested in the result of the encoder when contaminated with an energy-bounded perturbation, namely

$$\varphi_{\mathbf{D}}(\mathbf{x}_0 + \mathbf{v}) = \arg\min_{\mathbf{z}} \frac{1}{2}\|(\mathbf{x}_0 + \mathbf{v}) - \mathbf{D}\mathbf{z}\|_2^2 + \lambda\|\mathbf{z}\|_1, \quad (31)$$

where $\mathbf{v} \in \Delta_\nu = \{\mathbf{v} : \|\mathbf{v}\|_2 \leq \nu < 1\}$. We will often denote $\mathbf{x} = \mathbf{x}_0 + \mathbf{v}$ for simplicity. Also, note that there exist natural bounds for the penalty parameter, $0 \leq \lambda \leq (1 + \nu)$. The upper bound follows from the observation that as long as $\lambda > \|\mathbf{D}^T\mathbf{x}\|_\infty$, the solution of Eq.(31) is the zero vector. Since the columns of $\mathbf{D}$ are normalized, $\|\mathbf{D}^T(\mathbf{x}_0 + \mathbf{v})\|_\infty \leq \|\mathbf{x}_0 + \mathbf{v}\|_2 \leq 1 + \nu$.

Recall that from optimality conditions of Lasso, the solution $\varphi_{\mathbf{D}}(\mathbf{x})$ satisfies

$$|\mathbf{D}_i^T(\mathbf{x} - \mathbf{D}\varphi_{\mathbf{D}}(\mathbf{x}))| = \lambda \quad \text{if} \quad [\varphi_{\mathbf{D}}(\mathbf{x})]_i \neq 0 \quad (32)$$

$$|\mathbf{D}_i^T(\mathbf{x} - \mathbf{D}\varphi_{\mathbf{D}}(\mathbf{x}))| \leq \lambda \quad \text{if} \quad [\varphi_{\mathbf{D}}(\mathbf{x})]_i = 0. \quad (33)$$

Lastly, recall that the encoder gap assumption ($\tau_s \geq \tau_* > 0$) implies that there exist a set of inactive $(p - s)$ atoms $\mathcal{I}$ so that

$$|\mathbf{D}_i^T(\mathbf{x} - \mathbf{D}\varphi_\mathbf{D}(\mathbf{x}))| < \lambda - \tau_s$$

for all $i \in \mathcal{I}$.

Let us now formalize a few properties on the solution of the Lasso solution that will be used throughout.

**Remark B.2.** *For the setting above, we have that*

*a)* $\|(\mathbf{x}_0 + \mathbf{v}) - \mathbf{D}\varphi_\mathbf{D}(\mathbf{x}_0 + \mathbf{v})\|_2 \leq (1 + \nu)$,

*b)* $\|\varphi_\mathbf{D}(\mathbf{x}_0 + \mathbf{v})\|_2 \leq (1 + \nu)^2/(2\lambda)$,

*c)* $\|\mathbf{D}\varphi_\mathbf{D}(\mathbf{x}_0 + \mathbf{v})\|_2 \leq (1 + \nu)$,

*Proof.* Remarks $a)$ and $b)$ can be shown by noting that, by definition of the encoder,

$$\frac{1}{2}\|(\mathbf{x}_0 + \mathbf{v}) - \mathbf{D}\varphi_\mathbf{D}(\mathbf{x}_0 + \mathbf{v})\|_2^2 + \lambda\|\varphi_\mathbf{D}(\mathbf{x}_0 + \mathbf{v})\|_1 \leq \frac{1}{2}\|(\mathbf{x}_0 + \mathbf{v})\|_2^2 \leq \frac{1}{2}(1 + \nu)^2. \quad (34)$$

The above follows since the LHS is the minimum function value, attained precisely $\varphi_\mathbf{D}(\mathbf{x}_0 + \mathbf{v})$, whereas $\frac{1}{2}\|(\mathbf{x}_0 + \mathbf{v})\|_2^2$ is the function value for the alternative choice of $\mathbf{z} = 0$. The right-most inequality follows from the triangle inequality on $\|\mathbf{x}_0 + \mathbf{v}\|_2$.

For remark $c)$, denote $\mathbf{x} = \mathbf{x}_0 + \mathbf{v}$, and note that the minimizer of the above optimization problem satisfies (as follows from optimality of the minimizer [Mehta and Gray, 2013, Lemma 13 of Supplementary])

$$\frac{1}{2}\|\mathbf{x} - \mathbf{D}\varphi_\mathbf{D}(\mathbf{x})\|_2^2 + \lambda\|\varphi_\mathbf{D}(\mathbf{x})\|_1 = \frac{1}{2}\|\mathbf{x}\|_2^2 - \frac{1}{2}\|\mathbf{D}\varphi_\mathbf{D}(\mathbf{x})\|_2^2. \quad (35)$$

We expand the LHS and obtain a lower bound by Cauchy-Schwarz (and dropping the $\ell_1$ term)

$$\frac{1}{2}\|\mathbf{x}\| + \frac{1}{2}\|\mathbf{D}\varphi_\mathbf{D}(\mathbf{x})\|_2^2 - \mathbf{x}^T\mathbf{D}\varphi_\mathbf{D}(\mathbf{x}) + \lambda\|\varphi_\mathbf{D}(\mathbf{x})\|_1 \geq \frac{1}{2}\|\mathbf{x}\| + \frac{1}{2}\|\mathbf{D}\varphi_\mathbf{D}(\mathbf{x})\|_2^2 - \|\mathbf{x}\|_2\|\mathbf{D}\varphi_\mathbf{D}(\mathbf{x})\|_2 \tag{36}$$

$$\geq \frac{1}{2}\|\mathbf{x}\| + \frac{1}{2}\|\mathbf{D}\varphi_\mathbf{D}(\mathbf{x})\|_2^2 - (1 + \nu)\|\mathbf{D}\varphi_\mathbf{D}(\mathbf{x})\|_2 \tag{37}$$

Thus, together with (35), we have that $\|\mathbf{D}\varphi_\mathbf{D}(\mathbf{x} + \mathbf{v})\|_2 \leq (1 + \nu)$. □

**Lemma B.3.** *If $\|\mathbf{D} - \mathbf{D}'\| \leq \epsilon \leq 2\lambda/(1 + \nu)^2$, then*

$$\max_{\mathbf{v} \in \Delta_\nu} \left|\|\mathbf{D}\varphi_\mathbf{D}(\mathbf{x}_0 + \mathbf{v})\|_2^2 - \|\mathbf{D}'\varphi_{\mathbf{D}'}(\mathbf{x}_0 + \mathbf{v})\|_2^2\right| \leq \frac{5\epsilon}{2\lambda}(1 + \nu)^2. \quad (38)$$

The proof mimics that in [Mehta and Gray, 2013, Lemma 10-11], though accommodating for the adversarial perturbation. We include it here for completeness. Note that the above assumption on $\epsilon \leq 2\lambda/(1 + \nu)^2$ is mild, and it will hold under the setting of later lemmata.

*Proof.* Denote $\mathbf{x} = \mathbf{x}_0 + \mathbf{v}$. Let us further denote the optimal value attained by the encoders with one and other model as

$$v_\mathbf{D}^* = \min_z \frac{1}{2}\|\mathbf{x} - \mathbf{D}\mathbf{z}\|_2^2 + \lambda\|\mathbf{z}\|_1,$$

$$v_{\mathbf{D}'}^* = \min_z \frac{1}{2}\|\mathbf{x} - \mathbf{D}'\mathbf{z}\|_2^2 + \lambda\|\mathbf{z}\|_1.$$

Then, since this cost is only increased if using a different representation, we have that:

$$v_{\mathbf{D}'}^* \leq \frac{1}{2}\|\mathbf{x} - \mathbf{D}'\varphi_{\mathbf{D}}(\mathbf{x})\|_2^2 + \lambda\|\varphi_{\mathbf{D}}(\mathbf{x})\|_1 \tag{39}$$

$$= \frac{1}{2}\|\mathbf{x} - \mathbf{D}'\varphi_{\mathbf{D}}(\mathbf{x}) + (\mathbf{D} - \mathbf{D})\varphi_{\mathbf{D}}(\mathbf{x})\|_2^2 + \lambda\|\varphi_{\mathbf{D}}(\mathbf{x})\|_1 \tag{40}$$

$$\leq \frac{1}{2}\|\mathbf{x} - \mathbf{D}\varphi_{\mathbf{D}}(\mathbf{x})\|_2^2 + |\langle \mathbf{x} - \mathbf{D}\varphi_{\mathbf{D}}(\mathbf{x}), (\mathbf{D} - \mathbf{D}')\varphi_{\mathbf{D}}(\mathbf{x})\rangle| + \frac{1}{2}\|(\mathbf{D} - \mathbf{D}')\varphi_{\mathbf{D}}(\mathbf{x})\|_2^2 + \dots \tag{41}$$

$$\dots + \lambda\|\varphi_{\mathbf{D}}(\mathbf{x})\|_1 \tag{42}$$

$$\leq v_{\mathbf{D}}^* + \|\mathbf{x} - \mathbf{D}\varphi_{\mathbf{D}}(\mathbf{x})\|_2 \|\mathbf{D} - \mathbf{D}'\|_2 \|\varphi_{\mathbf{D}}(\mathbf{x})\|_2 + \frac{1}{2}\left(\|\mathbf{D} - \mathbf{D}'\|_2 \|\varphi_{\mathbf{D}}(\mathbf{x})\|_2\right)^2 \quad \text{by C.Swz.} \tag{43}$$

$$\leq v_{\mathbf{D}}^* + (1 + \nu)\epsilon \frac{(1 + \nu)^2}{2\lambda} + \frac{1}{2}\left(\frac{\epsilon(1 + \nu)^2}{2\lambda}\right)^2 \quad \text{by Remark B.2} \tag{44}$$

We further simplify the expression above by noting that $\nu < 1$ and that $\frac{\epsilon(1+\nu)^2}{2\lambda} \leq 1$ by assumption, obtaining

$$v_{\mathbf{D}'}^* \leq v_{\mathbf{D}}^* + \frac{5\epsilon}{4\lambda}(1 + \nu)^2. \tag{45}$$

Thus, from this (and a symmetric argument) follows that

$$|v_{\mathbf{D}'}^* - v_{\mathbf{D}}^*| \leq \frac{5\epsilon}{4\lambda}(1 + \nu)^2. \tag{46}$$

Lastly, recall from Eq. (35) that $v_{\mathbf{D}}^* = \frac{1}{2}\|\mathbf{x}\|_2^2 - \|\mathbf{D}\varphi_{\mathbf{D}}(\mathbf{x})\|_2^2$. Thus, using this expression for $v_{\mathbf{D}}^*$ and $v_{\mathbf{D}'}^*$ above, we get

$$\left|\|\mathbf{D}\varphi_{\mathbf{D}}(\mathbf{x})\|_2^2 - \|\mathbf{D}'\varphi_{\mathbf{D}'}(\mathbf{x})\|_2^2\right| \leq 2\left|v_{\mathbf{D}'}^* - v_{\mathbf{D}}^*\right| \leq \frac{5\epsilon}{2\lambda}(1 + \nu)^2. \tag{47}$$

□

We now show that if the dictionaries are close, then the reconstructions from one and other encoded representation are not too far either.

**Lemma B.4.** *If* $\|\mathbf{D} - \tilde{\mathbf{D}}\| \leq \epsilon \leq 2\lambda/(1 + \nu)^2$, *then*

$$\max_{\mathbf{v} \in \Delta_\nu} \|\mathbf{D}\varphi_{\mathbf{D}}(\mathbf{x}_0 + \mathbf{v}) - \mathbf{D}\varphi_{\tilde{\mathbf{D}}}(\mathbf{x}_0 + \mathbf{v})\|_2^2 \leq \frac{25\epsilon}{\lambda}(1 + \nu)^2. \tag{48}$$

*Proof.* For simplicity, denote $\mathbf{x} = \mathbf{x}_0 + \mathbf{v}$, as well as $\boldsymbol{\alpha} = \varphi_{\mathbf{D}}(\mathbf{x})$ and $\tilde{\boldsymbol{\alpha}} = \varphi_{\tilde{\mathbf{D}}}(\mathbf{x})$, where $\|\mathbf{D} - \tilde{\mathbf{D}}\|_2 \le \epsilon$. We first upper bound $\left| \|\mathbf{D}\boldsymbol{\alpha}\|_2^2 - \|\mathbf{D}\tilde{\boldsymbol{\alpha}}\|_2^2 \right|$ by a sequence of algebraic manipulations:

$$\left| \|\mathbf{D}\boldsymbol{\alpha}\|_2^2 - \|\mathbf{D}\tilde{\boldsymbol{\alpha}}\|_2^2 \right| \le \left| \|\mathbf{D}\boldsymbol{\alpha}\|_2^2 - \|\tilde{\mathbf{D}}\tilde{\boldsymbol{\alpha}}\|_2^2 \right| + \left| \|\mathbf{D}\tilde{\boldsymbol{\alpha}}\|_2^2 - \|\tilde{\mathbf{D}}\tilde{\boldsymbol{\alpha}}\|_2^2 \right| \qquad \pm \|\tilde{\mathbf{D}}\tilde{\boldsymbol{\alpha}}\|_2^2, \text{ triang. inq.} \tag{49}$$

$$\text{Lemma B.3, } \pm\tilde{\mathbf{D}} \le \frac{5\epsilon}{2\lambda}(1 + \nu)^2 + \left| \langle \mathbf{D}\tilde{\boldsymbol{\alpha}}, (\mathbf{D} - \tilde{\mathbf{D}} + \tilde{\mathbf{D}})\tilde{\boldsymbol{\alpha}} \rangle - \langle \tilde{\mathbf{D}}\tilde{\boldsymbol{\alpha}}, \tilde{\mathbf{D}}\tilde{\boldsymbol{\alpha}} \rangle \right| \tag{50}$$

$$= \frac{5\epsilon}{2\lambda}(1 + \nu)^2 + \left| \langle \mathbf{D}\tilde{\boldsymbol{\alpha}}, (\mathbf{D} - \tilde{\mathbf{D}})\tilde{\boldsymbol{\alpha}} \rangle + \langle \mathbf{D}\tilde{\boldsymbol{\alpha}} - \tilde{\mathbf{D}}\tilde{\boldsymbol{\alpha}}, \tilde{\mathbf{D}}\tilde{\boldsymbol{\alpha}} \rangle \right| \tag{51}$$

$$\text{by } \pm\mathbf{D} \quad = \frac{5\epsilon}{2\lambda}(1 + \nu)^2 + \left| \langle \mathbf{D}\tilde{\boldsymbol{\alpha}}, (\mathbf{D} - \tilde{\mathbf{D}})\tilde{\boldsymbol{\alpha}} \rangle + \langle \mathbf{D}\tilde{\boldsymbol{\alpha}} - \tilde{\mathbf{D}}\tilde{\boldsymbol{\alpha}}, (\tilde{\mathbf{D}} - \mathbf{D} + \mathbf{D})\tilde{\boldsymbol{\alpha}} \rangle \right| \tag{52}$$

$$= \frac{5\epsilon}{2\lambda}(1 + \nu)^2 + \left| \langle \mathbf{D}\tilde{\boldsymbol{\alpha}}, (\mathbf{D} - \tilde{\mathbf{D}})\tilde{\boldsymbol{\alpha}} \rangle + \langle (\mathbf{D} - \tilde{\mathbf{D}})\tilde{\boldsymbol{\alpha}}, \mathbf{D}\tilde{\boldsymbol{\alpha}} \rangle - \langle (\mathbf{D} - \tilde{\mathbf{D}})\tilde{\boldsymbol{\alpha}}, (\mathbf{D} - \tilde{\mathbf{D}})\tilde{\boldsymbol{\alpha}} \rangle \right| \tag{53}$$

$$\le \frac{5\epsilon}{2\lambda}(1 + \nu)^2 + 2 \left| \langle \mathbf{D}\tilde{\boldsymbol{\alpha}}, (\mathbf{D} - \tilde{\mathbf{D}})\tilde{\boldsymbol{\alpha}} \rangle \right| \qquad \text{by dropping } -\|(\mathbf{D} - \tilde{\mathbf{D}})\tilde{\boldsymbol{\alpha}}\|_2^2 \tag{54}$$

$$\le \frac{5\epsilon}{2\lambda}(1 + \nu)^2 + 2\|\mathbf{D}\tilde{\boldsymbol{\alpha}}\|_2 \|\mathbf{D} - \tilde{\mathbf{D}}\|_2 \|\tilde{\boldsymbol{\alpha}}\|_2 \qquad \text{by C.S. and operator norm} \tag{55}$$

$$\le \frac{5\epsilon}{2\lambda}(1 + \nu)^2 + 2\frac{\epsilon}{2\lambda}(1 + \nu)^2 \|\mathbf{D}\tilde{\boldsymbol{\alpha}}\|_2 \qquad \text{by Remark B.2} \tag{56}$$

The term $\|\mathbf{D}\tilde{\boldsymbol{\alpha}}\|_2$ cannot be directly bounded via Remark B.2 because $\tilde{\boldsymbol{\alpha}}$ is the representation computed via $\tilde{\mathbf{D}}$ (not $\mathbf{D}$). Instead, by letting $\Delta = \mathbf{D} - \tilde{\mathbf{D}}$, we can simplify the above bound as $\|\mathbf{D}\tilde{\boldsymbol{\alpha}}\|_2 \le \|\tilde{\mathbf{D}}\tilde{\boldsymbol{\alpha}}\|_2 + \|\Delta\tilde{\boldsymbol{\alpha}}\|_2 \le (1 + \nu) + \epsilon(1 + \nu)^2/(2\lambda) \le 2 + 1 = 3$. Then, resuming above,

$$\left| \|\mathbf{D}\boldsymbol{\alpha}\|_2^2 - \|\mathbf{D}\tilde{\boldsymbol{\alpha}}\|_2^2 \right| \le \frac{5\epsilon}{2\lambda}(1 + \nu)^2 + \frac{6\epsilon}{2\lambda}(1 + \nu)^2 = \frac{11\epsilon}{2\lambda}(1 + \nu)^2. \tag{57}$$

Now, by definition of $\boldsymbol{\alpha}$ (as the minimizer of the Lasso problem), we have that

$$v_{\mathbf{D}}^*(\mathbf{x}) = \frac{1}{2}\|\mathbf{x} - \mathbf{D}\boldsymbol{\alpha}\|_2^2 + \lambda\|\boldsymbol{\alpha}\|_1 \le \frac{1}{2}\left\| \mathbf{x} - \mathbf{D}\left(\frac{\boldsymbol{\alpha} + \tilde{\boldsymbol{\alpha}}}{2}\right) \right\|_2^2 + \lambda\left\| \frac{\boldsymbol{\alpha} + \tilde{\boldsymbol{\alpha}}}{2} \right\|_1. \tag{58}$$

We now expand the RHS above through the same algebraic manipulations:

$$v_{\mathbf{D}}^*(\mathbf{x}) \le \frac{1}{2}\left\| \mathbf{x} - \mathbf{D}\left(\frac{\boldsymbol{\alpha} + \tilde{\boldsymbol{\alpha}}}{2}\right) \right\|_2^2 + \lambda\left\| \frac{\boldsymbol{\alpha} + \tilde{\boldsymbol{\alpha}}}{2} \right\|_1 \tag{59}$$

$$= \frac{1}{2}\left( \|\mathbf{x}\|_2^2 - \langle \mathbf{x}, (\mathbf{D}\boldsymbol{\alpha} + \mathbf{D}\tilde{\boldsymbol{\alpha}}) \rangle + \frac{1}{4}\|\mathbf{D}\boldsymbol{\alpha} + \mathbf{D}\tilde{\boldsymbol{\alpha}}\|_2^2 \right) + \lambda\left\| \frac{\boldsymbol{\alpha} + \tilde{\boldsymbol{\alpha}}}{2} \right\|_1 \tag{60}$$

$$= \frac{1}{2}\|\mathbf{x}\|_2^2 - \frac{1}{2}\langle \mathbf{x}, \mathbf{D}\boldsymbol{\alpha} \rangle - \frac{1}{2}\langle \mathbf{x}, \mathbf{D}\tilde{\boldsymbol{\alpha}} \rangle + \frac{1}{8}\left( \|\mathbf{D}\boldsymbol{\alpha}\|_2^2 + \|\mathbf{D}\tilde{\boldsymbol{\alpha}}\|_2^2 + 2\langle \mathbf{D}\boldsymbol{\alpha}, \mathbf{D}\tilde{\boldsymbol{\alpha}} \rangle \right) + \lambda\left\| \frac{\boldsymbol{\alpha} + \tilde{\boldsymbol{\alpha}}}{2} \right\|_1 \tag{61}$$

$$\le \frac{1}{2}\|\mathbf{x}\|_2^2 - \frac{1}{2}\langle \mathbf{x}, \mathbf{D}\boldsymbol{\alpha} \rangle - \frac{1}{2}\langle \mathbf{x}, \mathbf{D}\tilde{\boldsymbol{\alpha}} \rangle + \frac{1}{4}\|\mathbf{D}\boldsymbol{\alpha}\|_2^2 + \frac{1}{4}\langle \mathbf{D}\boldsymbol{\alpha}, \mathbf{D}\tilde{\boldsymbol{\alpha}} \rangle + \dots \tag{62}$$

$$\dots + \frac{\lambda}{2}\|\boldsymbol{\alpha}\|_1 + \frac{\lambda}{2}\|\tilde{\boldsymbol{\alpha}}\|_1 + \frac{11}{16}\frac{\epsilon}{\lambda}(1 + \nu)^2, \tag{63}$$

where the last step follows by adding and subtracting $\|\mathbf{D}\boldsymbol{\alpha}\|_2^2$ and employing the bound obtained above in (57). Now, from Eq. (35), we can write

$$\lambda\|\boldsymbol{\alpha}\|_1 = \langle \mathbf{x} - \mathbf{D}\boldsymbol{\alpha}, \mathbf{D}\boldsymbol{\alpha} \rangle = \langle \mathbf{x}, \mathbf{D}\boldsymbol{\alpha} \rangle - \|\mathbf{D}\boldsymbol{\alpha}\|_2^2. \tag{64}$$

The expression for $\|\tilde{\boldsymbol{\alpha}}\|_1$ is expanded similarly but then upper bounded via Lemma B.3 by adding and subtracting $\|\mathbf{D}\boldsymbol{\alpha}\|_2^2$:

$$\lambda\|\tilde{\boldsymbol{\alpha}}\|_1 = \langle \mathbf{x} - \tilde{\mathbf{D}}\tilde{\boldsymbol{\alpha}}, \tilde{\mathbf{D}}\tilde{\boldsymbol{\alpha}}\rangle \leq \langle \mathbf{x}, \tilde{\mathbf{D}}\tilde{\boldsymbol{\alpha}}\rangle - \|\mathbf{D}\boldsymbol{\alpha}\|_2^2 + \frac{5\epsilon}{2\lambda}(1+\nu)^2 \tag{65}$$

$$= \langle \mathbf{x}, \mathbf{D}\tilde{\boldsymbol{\alpha}}\rangle + \langle \mathbf{x}, (\tilde{\mathbf{D}} - \mathbf{D})\tilde{\boldsymbol{\alpha}}\rangle - \|\mathbf{D}\boldsymbol{\alpha}\|_2^2 + \frac{5\epsilon}{2\lambda}(1+\nu)^2 \qquad \text{by } \pm\mathbf{D} \tag{66}$$

$$\text{by C.S.} \leq \langle \mathbf{x}, \mathbf{D}\tilde{\boldsymbol{\alpha}}\rangle + \|\mathbf{x}\|_2\|\tilde{\mathbf{D}} - \mathbf{D}\|_2\|\tilde{\boldsymbol{\alpha}}\|_2 - \|\mathbf{D}\boldsymbol{\alpha}\|_2^2 + \frac{5\epsilon}{2\lambda}(1+\nu)^2 \tag{67}$$

$$\leq \langle \mathbf{x}, \mathbf{D}\tilde{\boldsymbol{\alpha}}\rangle + (1+\nu)\epsilon\frac{(1+\nu)^2}{2\lambda} - \|\mathbf{D}\boldsymbol{\alpha}\|_2^2 + \frac{5\epsilon}{2\lambda}(1+\nu)^2 \tag{68}$$

$$\leq \langle \mathbf{x}, \mathbf{D}\tilde{\boldsymbol{\alpha}}\rangle - \|\mathbf{D}\boldsymbol{\alpha}\|_2^2 + \frac{7\epsilon}{2\lambda}(1+\nu)^2. \tag{69}$$

Thus, we can now upper bound the expression for $v_{\mathbf{D}}^*$ in Eq. (35) by combining Eq. (62), (64) and (69) as follows. From Eq. (35) and the bound in Eq. (62) we get:

$$v_{\mathbf{D}}^* = \frac{1}{2}\|\mathbf{x}\|_2^2 - \frac{1}{2}\|\mathbf{D}\boldsymbol{\alpha}\|_2^2 \tag{70}$$

$$\leq \frac{1}{2}\|\mathbf{x}\|_2^2 - \frac{1}{2}\langle \mathbf{x}, \mathbf{D}\boldsymbol{\alpha}\rangle - \frac{1}{2}\langle \mathbf{x}, \mathbf{D}\tilde{\boldsymbol{\alpha}}\rangle + \frac{1}{4}\|\mathbf{D}\boldsymbol{\alpha}\|_2^2 + \frac{1}{4}\langle \mathbf{D}\boldsymbol{\alpha}, \mathbf{D}\tilde{\boldsymbol{\alpha}}\rangle + \dots \dots \tag{71}$$

$$\dots + \frac{\lambda}{2}\|\boldsymbol{\alpha}\|_1 + \frac{\lambda}{2}\|\tilde{\boldsymbol{\alpha}}\|_1 + \frac{11}{16}\frac{\epsilon}{\lambda}(1+\nu)^2. \tag{72}$$

Replacing now the expression for $\lambda\|\boldsymbol{\alpha}\|_1$ from (64) and the upper bound for $\lambda\|\tilde{\boldsymbol{\alpha}}\|_1$ from (69):

$$v_{\mathbf{D}}^* = \frac{1}{2}\|\mathbf{x}\|_2^2 - \frac{1}{2}\|\mathbf{D}\boldsymbol{\alpha}\|_2^2 \leq \frac{1}{2}\|\mathbf{x}\|_2^2 - \frac{3}{4}\|\mathbf{D}\boldsymbol{\alpha}\|_2^2 + \frac{1}{4}\langle \mathbf{D}\boldsymbol{\alpha}, \mathbf{D}\tilde{\boldsymbol{\alpha}}\rangle + \frac{39}{16}\frac{\epsilon}{\lambda}(1+\nu)^2, \tag{73}$$

which leads to

$$-\frac{1}{2}\|\mathbf{D}\boldsymbol{\alpha}\|_2^2 \leq -\frac{3}{4}\|\mathbf{D}\boldsymbol{\alpha}\|_2^2 + \frac{1}{4}\langle \mathbf{D}\boldsymbol{\alpha}, \mathbf{D}\tilde{\boldsymbol{\alpha}}\rangle + \frac{39}{16}\frac{\epsilon}{\lambda}(1+\nu)^2, \tag{74}$$

and so

$$\|\mathbf{D}\boldsymbol{\alpha}\|_2^2 \leq \langle \mathbf{D}\boldsymbol{\alpha}, \mathbf{D}\tilde{\boldsymbol{\alpha}}\rangle + \frac{39}{4}\frac{\epsilon}{\lambda}(1+\nu)^2. \tag{75}$$

Finally, with this expression we can now bound the distance

$$\|\mathbf{D}\boldsymbol{\alpha} - \mathbf{D}\tilde{\boldsymbol{\alpha}}\|_2^2 = \|\mathbf{D}\boldsymbol{\alpha}\|_2^2 + \|\mathbf{D}\tilde{\boldsymbol{\alpha}}\|_2^2 - 2\langle \mathbf{D}\boldsymbol{\alpha}, \mathbf{D}\tilde{\boldsymbol{\alpha}}\rangle \tag{76}$$

$$\leq \|\mathbf{D}\boldsymbol{\alpha}\|_2^2 + \|\mathbf{D}\tilde{\boldsymbol{\alpha}}\|_2^2 - 2\|\mathbf{D}\boldsymbol{\alpha}\|_2^2 + \frac{39}{2}\frac{\epsilon}{\lambda}(1+\nu)^2 \qquad \text{by the expression above} \tag{77}$$

$$= \|\mathbf{D}\tilde{\boldsymbol{\alpha}}\|_2^2 - \|\mathbf{D}\boldsymbol{\alpha}\|_2^2 + \frac{39}{2}\frac{\epsilon}{\lambda}(1+\nu)^2 \tag{78}$$

$$\leq 25\frac{\epsilon}{\lambda}(1+\nu)^2, \tag{79}$$

where the last inequality follows from Eq. (57).

$$\square$$

We will now show that if the dictionaries are close enough and if the solution of one of them was at most $s$-sparse and has a positive encoder gap, then the solution with the perturbed model retains the sparsity. This is inspired by the result in [Mehta and Gray, 2013]. However, because of the adversarial perturbation, extra work is required to provide a condition on the *unperturbed* gap, i.e. that which will withstand adversarial energy-bounded perturbation. The following Lemma will be necessary to show the result:

**Lemma B.5.** *If $\|\mathbf{v}\|_2 \leq \nu$, then*

$$\|\mathbf{D}\varphi_{\mathbf{D}}(\mathbf{x}) - \mathbf{D}\varphi_{\mathbf{D}}(\mathbf{x} + \mathbf{v})\|_2^2 \leq \nu^2. \tag{80}$$

Let us postpone the proof of this result for later. We are now ready to state and prove the preservation of sparsity result:

**Lemma B.6.** *(Preservation of sparsity under model deviation and adversarial perturbations) Consider $\varphi_{\mathbf{D}}(\mathbf{x}_0 + \mathbf{v})$, for $\|\mathbf{v}\|_2 \leq \nu$, and an alternative dictionary $\tilde{\mathbf{D}}$ so that $\|\mathbf{D} - \tilde{\mathbf{D}}\|_2 \leq \epsilon \leq 2\lambda/(1 + \nu)^2$. If there exist a set of inactive $(p - s)$ atoms $\mathcal{I}$ so that*

$$|\mathbf{D}_i^T(\mathbf{x}_0 - \mathbf{D}\varphi_{\mathbf{D}}(\mathbf{x}_0))| < \lambda - \tau_s \tag{81}$$

*for all $i \in \mathcal{I}$, and*

$$\tau_s > 2\nu + \sqrt{\epsilon}\left(\sqrt{\frac{25}{\lambda}}(1 + \nu) + 2\left(\frac{(1 + \nu)}{\lambda} + 1\right)\right), \tag{82}$$

*then $[\varphi_{\tilde{\mathbf{D}}}(\mathbf{x}_0 + \mathbf{v})]_i = 0 \; \forall i \in \mathcal{I}$, where (reminder)*

$$\varphi_{\tilde{\mathbf{D}}}(\mathbf{x}_0 + \mathbf{v}) = \arg\min_{\mathbf{z}} \frac{1}{2}\|(\mathbf{x}_0 + \mathbf{v}) - \tilde{\mathbf{D}}\mathbf{z}\|_2^2 + \lambda\|\mathbf{z}\|_1. \tag{83}$$

*Proof.* Let $\mathbf{x} = \mathbf{x}_0 + \mathbf{v}$, as well as $\boldsymbol{\alpha} = \varphi_{\mathbf{D}}(\mathbf{x})$, $\tilde{\boldsymbol{\alpha}} = \varphi_{\tilde{\mathbf{D}}}(\mathbf{x})$, and let $\mathcal{I}$ be the set of $(p - s)$ inactive atoms with positive gap $\tau_s$.

In order for the inactive set of atoms $\mathcal{I}$ to remain inactive, we need to show that $\forall i \in \mathcal{I}$,

$$\left|\langle \tilde{\mathbf{D}}_i, \mathbf{x} - \tilde{\mathbf{D}}\tilde{\boldsymbol{\alpha}}\rangle\right| < \lambda.$$

Consider the following upper bound to the LHS above:

$$\left|\langle \tilde{\mathbf{D}}_i, \mathbf{x} - \tilde{\mathbf{D}}\tilde{\boldsymbol{\alpha}}\rangle\right| \leq \left|\langle \mathbf{D}_i, \mathbf{x} - \tilde{\mathbf{D}}\tilde{\boldsymbol{\alpha}}\rangle\right| + \|\tilde{\mathbf{D}}_i - \mathbf{D}_i\|_2\|\mathbf{x} - \tilde{\mathbf{D}}\tilde{\boldsymbol{\alpha}}\|_2 \quad \text{by } \pm\mathbf{D}_i \text{ and C.S.} \tag{84}$$

$$\leq \left|\langle \mathbf{D}_i, \mathbf{x} - \tilde{\mathbf{D}}\tilde{\boldsymbol{\alpha}}\rangle\right| + \epsilon(1 + \nu) \tag{85}$$

$$\leq |\langle \mathbf{D}_i, \mathbf{x} - \mathbf{D}\tilde{\boldsymbol{\alpha}}\rangle| + \left|\langle \mathbf{D}_i, (\tilde{\mathbf{D}} - \mathbf{D})\tilde{\boldsymbol{\alpha}}\rangle\right| + \epsilon(1 + \nu) \quad \text{by } \pm\mathbf{D} \text{ and triang ineq.} \tag{86}$$

$$\leq |\langle \mathbf{D}_i, \mathbf{x} - \mathbf{D}\tilde{\boldsymbol{\alpha}}\rangle| + \|\mathbf{D}_i\|_2\|\tilde{\mathbf{D}} - \mathbf{D}\|_2\|\tilde{\boldsymbol{\alpha}}\|_2 + \epsilon(1 + \nu) \tag{87}$$

$$\leq |\langle \mathbf{D}_i, \mathbf{x} - \mathbf{D}\tilde{\boldsymbol{\alpha}}\rangle| + \frac{\epsilon}{2\lambda}(1 + \nu)^2 + \epsilon(1 + \nu). \quad \text{by Remark B.2 and unit-norm columns} \tag{88}$$

Thus, it is sufficient to show that

$$|\langle \mathbf{D}_i, \mathbf{x} - \mathbf{D}\tilde{\boldsymbol{\alpha}}\rangle| < \lambda - \epsilon(1 + \nu)\left(\frac{(1 + \nu)}{2\lambda} + 1\right). \tag{89}$$

Let us now replace $\mathbf{x}$, $\boldsymbol{\alpha}$ and $\tilde{\boldsymbol{\alpha}}$ by their definitions and upper bound the left hand side above by using Lemma B.4 and Lemma B.5

$$|\langle \mathbf{D}_i, \mathbf{x} - \mathbf{D}\tilde{\boldsymbol{\alpha}}\rangle| = |\langle \mathbf{D}_i, (\mathbf{x}_0 + \mathbf{v}) - \mathbf{D}\varphi_{\tilde{\mathbf{D}}}(\mathbf{x}_0 + \mathbf{v})\rangle| \tag{90}$$

$$\leq \left|\langle \mathbf{D}_i, (\mathbf{x}_0 + \mathbf{v}) - \mathbf{D}\varphi_{\mathbf{D}}(\mathbf{x}_0 + \mathbf{v})\right| + |\langle \mathbf{D}_i, \mathbf{D}\varphi_{\mathbf{D}}(\mathbf{x}_0 + \mathbf{v}) - \mathbf{D}\varphi_{\tilde{\mathbf{D}}}(\mathbf{x}_0 + \mathbf{v})| \quad \text{by } \pm\boldsymbol{\alpha} \tag{91}$$

$$\leq \overbrace{|\langle \mathbf{D}_i, \mathbf{x}_0 - \mathbf{D}\varphi_{\mathbf{D}}(\mathbf{x}_0 + \mathbf{v})| + \|\mathbf{D}_i\|_2\|\mathbf{v}\|_2} + \|\mathbf{D}_i\|_2\|\mathbf{D}\varphi_{\mathbf{D}}(\mathbf{x}_0 + \mathbf{v}) - \mathbf{D}\varphi_{\tilde{\mathbf{D}}}(\mathbf{x}_0 + \mathbf{v})\|_2 \tag{92}$$

$$\leq |\langle \mathbf{D}_i, \mathbf{x}_0 - \mathbf{D}\varphi_{\mathbf{D}}(\mathbf{x}_0 + \mathbf{v})| + \nu + \sqrt{\frac{25\epsilon}{\lambda}}(1 + \nu) \quad \text{by Lemma B.4} \tag{93}$$

$$\text{by } \pm\varphi_{\mathbf{D}}(\mathbf{x}_0) \quad \leq |\langle \mathbf{D}_i, \mathbf{x}_0 - \mathbf{D}\varphi_{\mathbf{D}}(\mathbf{x}_0)| + \|\mathbf{D}_i\|_2\|\mathbf{D}\varphi_{\mathbf{D}}(\mathbf{x}_0) - \mathbf{D}\varphi_{\mathbf{D}}(\mathbf{x}_0 + \mathbf{v})\|_2 + \nu + \sqrt{\frac{25\epsilon}{\lambda}}(1 + \nu) \tag{94}$$

$$\leq |\langle \mathbf{D}_i, \mathbf{x}_0 - \mathbf{D}\varphi_{\mathbf{D}}(\mathbf{x}_0)| + 2\nu + \sqrt{\frac{25\epsilon}{\lambda}}(1 + \nu) \quad \text{by Lemma B.5} \tag{95}$$

$$< \lambda - \tau_s + 2\nu + \sqrt{\frac{25\epsilon}{\lambda}}(1 + \nu) \tag{96}$$

where the last step follows from the assumption of the encoder gap in Eq. (81). Thus, merging with (89), we require

$$-\tau_s + 2\nu + \sqrt{\frac{25\epsilon}{\lambda}}(1+\nu) < -\epsilon(1+\nu)\left(\frac{(1+\nu)}{2\lambda}+1\right),$$

implying that as long as

$$\tau_s > 2\nu + \sqrt{\frac{25\epsilon}{\lambda}}(1+\nu) + \epsilon(1+\nu)\left(\frac{(1+\nu)}{2\lambda}+1\right) \tag{97}$$

the inactive set $\mathcal{I}$ remains inactive. For the sake of simplicity, we will make the above condition more stringent. Using the fact that $\nu < 1$ and that $\epsilon \le 1$. Thus,

$$\tau_s > 2\nu + \sqrt{\epsilon}\left(\sqrt{\frac{25}{\lambda}}(1+\nu) + 2\left(\frac{(1+\nu)}{2\lambda}+1\right)\right). \tag{98}$$

$\square$

The lemma above is central, as it guarantees that a sparsity of up to $s$ non-zeros is retained under model deviations and adversarial perturbations. Lemma B.3 now follows directly from the proof of Theorem 4 in [Mehta and Gray, 2013], albeit with the constants provided by Remark B.2 which account for the perturbation $\mathbf{v}$.

We owe the proof of Lemma B.5. This Lemma will also be instrumental in the proof of Theorem C.1. We now re-state it, and proceed to prove it.

**Lemma B.7.** *5.4 (Norm Stability under adversarial perturbations)*
*If $\|\mathbf{v}\|_2 \le \nu$, then*

$$\|\mathbf{D}\varphi_{\mathbf{D}}(\mathbf{x}) - \mathbf{D}\varphi_{\mathbf{D}}(\mathbf{x}+\mathbf{v})\|_2^2 \le \nu^2 \tag{99}$$

*Proof.* We will re-formulate the Lasso problem as an equivalent quadratic program, and then utilize optimality properties of its solution. Let us define the vector $\bar{\mathbf{z}} \in \mathbb{R}^{3p}$ such that $\bar{\mathbf{z}} = [\mathbf{z}, \mathbf{z}^+, \mathbf{z}^-]^T$, with $\mathbf{z}^+$ and $\mathbf{z}^-$ containing all positive and negative elements in $\mathbf{z}$, respectively. Define then the following quadratic cost

$$Q(\bar{\mathbf{z}}, \mathbf{x}) = \frac{1}{2}\bar{\mathbf{z}}^T \begin{bmatrix} \mathbf{D}^T\mathbf{D} & \mathbf{0}_{p\times 2p} \\ \mathbf{0}_{2p\times p} & \mathbf{0}_{2p\times 2p} \end{bmatrix} \bar{\mathbf{z}} - \bar{\mathbf{z}}^T \begin{bmatrix} \mathbf{D}^T \\ \mathbf{0}_{2p\times d} \end{bmatrix} \mathbf{x} + \lambda[\mathbf{0}_p^T, \mathbf{1}_{2p}^T]\bar{\mathbf{z}}. \tag{100}$$

With this definition, the Lasso problem can be re-formulated as the following quadratic program:

$$\min_{\bar{\mathbf{z}}\in\mathbb{R}^{3p}} Q(\bar{\mathbf{z}}, \mathbf{x}) \quad \text{subject to} \quad \bar{\mathbf{z}} \in \mathcal{K} = \{\bar{\mathbf{z}} : \mathbf{z} = \mathbf{z}^+ - \mathbf{z}^-; \; \mathbf{z}^+, \mathbf{z}^- \ge 0\}. \tag{101}$$

Let us denote $Q(\bar{\mathbf{z}}) = Q(\bar{\mathbf{z}}, \mathbf{x}_0)$ and $\tilde{Q}(\bar{\mathbf{z}}) = Q(\bar{\mathbf{z}}, \mathbf{x}_0+\mathbf{v})$ for short, and $\boldsymbol{\beta}$ and $\tilde{\boldsymbol{\beta}}$ as the solution to the quadratic program with $Q(\bar{\mathbf{z}})$ and $\tilde{Q}(\bar{\mathbf{z}})$, respectively. Moreover, denote $\boldsymbol{\alpha} = \varphi_{\mathbf{D}}(\mathbf{x}_0)$ and $\tilde{\boldsymbol{\alpha}} = \varphi_{\mathbf{D}}(\mathbf{x}_0+\mathbf{v})$. With this notation, note that

$$\boldsymbol{\beta} = \begin{bmatrix} \boldsymbol{\alpha} \\ \boldsymbol{\alpha}^+ \\ \boldsymbol{\alpha}^- \end{bmatrix} \quad \text{and} \quad \tilde{\boldsymbol{\beta}} = \begin{bmatrix} \tilde{\boldsymbol{\alpha}} \\ \tilde{\boldsymbol{\alpha}}^+ \\ \tilde{\boldsymbol{\alpha}}^- \end{bmatrix}.$$

Note that the above problem in (101) is the minimization of a convex differentiable function over a convex set and therefore, for every $\bar{\mathbf{z}} \in \mathcal{K}$,

$$(\bar{\mathbf{z}} - \boldsymbol{\beta})^T \nabla_{\mathbf{z}} Q(\boldsymbol{\beta}) \ge 0 \tag{102a}$$
$$(\bar{\mathbf{z}} - \tilde{\boldsymbol{\beta}})^T \nabla_{\mathbf{z}} \tilde{Q}(\tilde{\boldsymbol{\beta}}) \ge 0. \tag{102b}$$

This gradient can be written as

$$\nabla Q(\bar{\mathbf{z}}) = \begin{bmatrix} \mathbf{D}^T\mathbf{D} & \mathbf{0}_{p\times 2p} \\ \mathbf{0}_{2p\times p} & \mathbf{0}_{2p\times 2p} \end{bmatrix} \bar{\mathbf{z}} - \begin{bmatrix} \mathbf{D}^T \\ \mathbf{0}_{2p\times d} \end{bmatrix} \mathbf{x} + \lambda \begin{bmatrix} \mathbf{0}_p \\ \mathbf{1}_{2p} \end{bmatrix}. \tag{103}$$

Now, choosing $\tilde{\boldsymbol{\beta}}$ as $\bar{\mathbf{z}}$ in (102a), $\boldsymbol{\beta}$ as $\bar{\mathbf{z}}$ in (102b) and subtracting one from the other, we get

$$(\tilde{\boldsymbol{\beta}} - \boldsymbol{\beta})^T \left( \nabla Q(\boldsymbol{\beta}) - \nabla \tilde{Q}(\tilde{\boldsymbol{\beta}}) \right) \geq 0 \tag{104}$$

which after employing the definitions for $\boldsymbol{\beta}, \tilde{\boldsymbol{\beta}}, \nabla Q$ and $\nabla \tilde{Q}$ results in

$$(\tilde{\boldsymbol{\alpha}} - \boldsymbol{\alpha})^T (\mathbf{D}^T \mathbf{D}(\boldsymbol{\alpha} - \tilde{\boldsymbol{\alpha}}) + \mathbf{D}^T \mathbf{v}) \geq 0. \tag{105}$$

The lemma is proven by finally expanding the above and employing Cauchy-Schwarz:

$$\|\mathbf{D}\tilde{\boldsymbol{\alpha}} - \mathbf{D}\boldsymbol{\alpha}\|_2^2 \leq (\tilde{\boldsymbol{\alpha}} - \boldsymbol{\alpha})^T \mathbf{D}^T \mathbf{v} \leq \|\mathbf{v}\|_2 \|\mathbf{D}\tilde{\boldsymbol{\alpha}} - \mathbf{D}\boldsymbol{\alpha}\|_2 \leq \nu \|\mathbf{D}\tilde{\boldsymbol{\alpha}} - \mathbf{D}\boldsymbol{\alpha}\|_2. \tag{106}$$

$\square$

## C  Robustness Certificate

**Theorem C.1** (Robustness certificate for binary predictive sparse coding). *Consider the predictor $f_{\mathbf{D},\mathbf{w}}(\mathbf{x})$, computed via $\varphi_{\mathbf{D}}(\mathbf{x})$ with an encoder gap of $\tau_s(\mathbf{x})$ and $\eta_s$-RIP dictionary $\mathbf{D}$. Then,*

$$sign(f_{\mathbf{D},\mathbf{w}}(\mathbf{x})) = sign(f_{\mathbf{D},\mathbf{w}}(\mathbf{x} + \mathbf{v})), \quad \forall \mathbf{v} : \|\mathbf{v}\|_2 \leq \nu \tag{107}$$

*so long as $\nu < \min\{ \tau_s(\mathbf{x})/2 , \rho_{\mathbf{x}}\sqrt{1 - \eta_s} \}$.*

We now proceed to prove Theorem C.1. We first must show that if there exist a positive encoder gap for a particular inactive set, this set will remain inactive under adversarial perturbations. This follows as a particular case of Lemma B.6 with $\epsilon = 0$, i.e. when there is no difference between the dictionaries: $\|\mathbf{D} - \tilde{\mathbf{D}}\|_2 = 0$. We re-state it here for completeness in this simplified form.

**Corollary C.2.** *Consider $\varphi_{\mathbf{D}}(\mathbf{x}_0)$ and $\varphi_{\mathbf{D}}(\mathbf{x}_0 + \mathbf{v})$, for $\|\mathbf{v}\|_2 \leq \nu$. If there exist a set of inactive $(p - s)$ atoms $\mathcal{I}$ in $\varphi_{\mathbf{D}}(\mathbf{x}_0)$ so that*

$$|\mathbf{D}_i^T (\mathbf{x}_0 - \mathbf{D}\varphi_{\mathbf{D}}(\mathbf{x}_0))| < \lambda - \tau_s \tag{108}$$

*for all $i \in \mathcal{I}$, and*

$$\tau_s > 2\nu, \tag{109}$$

*then $[\varphi_{\tilde{\mathbf{D}}}(\mathbf{x}_0 + \mathbf{v})]_i = 0 \; \forall i \in \mathcal{I}$.*

With this result, we now present a Lemma guaranteeing that the original and adversarially perturbed representation are not too far.

**Lemma 5.2** (Stability of representations under adversarial perturbations). *Let $\varphi_{\mathbf{D}}(\mathbf{x}_0)$ and $\varphi_{\mathbf{D}}(\mathbf{x}_0 + \mathbf{v})$, for $\|\mathbf{v}\|_2 \leq \nu$. If $\varphi_{\mathbf{D}}(\mathbf{x}_0)$ has an encoder gap $\tau_s > 2\nu$, and the dictionary is RIP with constant $\eta_s$, then*

$$\|\varphi_{\mathbf{D}}(\mathbf{x}_0) - \varphi_{\mathbf{D}}(\mathbf{x}_0 + \mathbf{v})\|_2 \leq \frac{\nu}{\sqrt{1 - \eta_s}}. \tag{110}$$

*Proof.* The proof of this result is now simple given our previous developments. On one hand, we have from Lemma B.7 that

$$\|\mathbf{D}\varphi_{\mathbf{D}}(\mathbf{x}_0) - \mathbf{D}\varphi_{\mathbf{D}}(\mathbf{x}_0 + \mathbf{v})\|_2^2 \leq \nu^2. \tag{111}$$

On the other hand, since $\varphi_{\mathbf{D}}(\mathbf{x}_0)$ has an encoder gap of $\tau_s > 2\nu$, there exist an inactive set of $(p - s)$ atoms that is retained in $\varphi_{\mathbf{D}}(\mathbf{x}_0 + \mathbf{v})$ by Corollary C.2. Thus, $\|\varphi_{\mathbf{D}}(\mathbf{x}_0) - \varphi_{\mathbf{D}}(\mathbf{x}_0 + \mathbf{v})\|_0 \leq s$. As a result, since $\mathbf{D}$ is $\eta_s$-RIP, we can write

$$\|\mathbf{D}\varphi_{\mathbf{D}}(\mathbf{x}_0) - \mathbf{D}\varphi_{\mathbf{D}}(\mathbf{x}_0 + \mathbf{v})\|_2^2 \geq (1 - \eta_s)\|\varphi_{\mathbf{D}}(\mathbf{x}_0) - \varphi_{\mathbf{D}}(\mathbf{x}_0 + \mathbf{v})\|_2^2. \tag{112}$$

Combining the lower and upper bounds proves the claim. $\square$

We are now ready to prove the result in Theorem C.1.

*Proof.* The proof is simple and inspired by the analysis in [Romano et al., 2019]. Recall that the hypothesis is implemented by $f_{\mathbf{D},\mathbf{w}}(\mathbf{x}) = \langle \mathbf{w}, \varphi_{\mathbf{D}}(\mathbf{x}) \rangle$. Since $\varphi_{\mathbf{D}}(\mathbf{x})$ has an encoder gap $\tau_s \geq 2\nu$, then it follows from the above Lemma 5.2 that

$$\|\varphi_{\mathbf{D}}(\mathbf{x}_0) - \varphi_{\mathbf{D}}(\mathbf{x}_0 + \mathbf{v})\|_2 \leq \frac{\nu}{\sqrt{1-\eta_s}}. \tag{113}$$

Without loss of generality, consider the case when $f_{\mathbf{D},\mathbf{w}}(\mathbf{x}) > 0$. Let us lower bound $f_{\mathbf{D},\mathbf{w}}(\mathbf{x} + \mathbf{v})$ as follows:

$$\langle \mathbf{w}, \varphi_{\mathbf{D}}(\mathbf{x} + \mathbf{v}) \rangle = \langle \mathbf{w}, \varphi_{\mathbf{D}}(\mathbf{x}) \rangle + \langle \mathbf{w}, \varphi_{\mathbf{D}}(\mathbf{x} + \mathbf{v}) - \varphi_{\mathbf{D}}(\mathbf{x}) \rangle \tag{114}$$

$$\geq \rho_{\mathbf{x}} \|\mathbf{w}\|_2 - |\langle \mathbf{w}, \varphi_{\mathbf{D}}(\mathbf{x} + \mathbf{v}) - \varphi_{\mathbf{D}}(\mathbf{x}) \rangle| \tag{115}$$

$$\geq \|\mathbf{w}\|_2 \left( \rho_{\mathbf{x}} - \frac{\nu}{\sqrt{1-\eta_s}} \right). \tag{116}$$

$$\tag{117}$$

Therefore, as long as $\rho_x > \frac{\nu}{\sqrt{1-\eta_s}}$ (and $\mathbf{w} \neq \mathbf{0}$), $sign(f_{\mathbf{D},\mathbf{w}}(\mathbf{x})) = sign(f_{\mathbf{D},\mathbf{w}}(\mathbf{x} + \mathbf{v}))$. $\square$

**Theorem 5.1** (Robustness Certificate for multiclass supervised sparse coding). *Let $\rho_x$ be the multiclass classifier margin of $f_{\mathbf{D},\mathbf{w}}(\mathbf{x})$, with $\rho_{\mathbf{x}} > 0$, composed of an encoder with gap of $\tau_s(\mathbf{x})$ and $\eta_s$-RIP dictionary $\mathbf{D}$. Furthermore, denote by $c_{\mathbf{W}} = \max_{i \neq j} \|\mathbf{W}_i - \mathbf{W}_j\|_2$ Then,*

$$\arg \max_{j \in [K]} [\mathbf{W}^T f_{\mathbf{D},\mathbf{w}}(\mathbf{x})] = \arg \max_{j \in [K]} [\mathbf{W}^T f_{\mathbf{D},\mathbf{w}}(\mathbf{x} + \mathbf{v})], \quad \forall \, \mathbf{v} : \|\mathbf{v}\|_2 \leq \nu \tag{118}$$

*so long as $\nu \leq \min\{\tau_s(\mathbf{x})/2, \rho_{\mathbf{x}}\sqrt{1-\eta_s}/c_{\mathbf{W}}\}$.*

*Proof.* Consider a sample with a positive multiclass margin:

$$\rho_{\mathbf{x}} = \mathbf{W}_y^T \varphi_{\mathbf{D}}(\mathbf{x}) - \max_{j \neq y} \mathbf{W}_j^T \varphi_{\mathbf{D}}(\mathbf{x}) > 0,$$

and let us lower-bound the margin on the perturbed input $f_{\mathbf{D},\mathbf{w}}(\mathbf{x} + \mathbf{v})$ as follows:

$$\rho_{\mathbf{x}+\mathbf{v}} = \mathbf{W}_y^T \varphi_{\mathbf{D}}(\mathbf{x} + \mathbf{v}) - \max_{j \neq y} \mathbf{W}_j^T \varphi_{\mathbf{D}}(\mathbf{x} + \mathbf{v}) \tag{119}$$

$$= \min_{j \neq y} \langle \mathbf{W}_y - \mathbf{W}_j, \varphi_{\mathbf{D}}(\mathbf{x} + \mathbf{v}) \rangle \tag{120}$$

$$\geq \min_{j \neq y} \langle \mathbf{W}_y - \mathbf{W}_j, \varphi_{\mathbf{D}}(\mathbf{x}) \rangle - \|\mathbf{W}_y - \mathbf{W}_j\|_2 \|\varphi_{\mathbf{D}}(\mathbf{x} + \mathbf{v}) - \varphi_{\mathbf{D}}(\mathbf{x})\|_2 \tag{121}$$

$$\geq \rho_{\mathbf{x}} - c_{\mathbf{W}} \, \nu / \sqrt{1-\eta_s}, \tag{122}$$

where the second-to-last inequality follows from hypothesis and Lemma 5.2. Therefore, as long as $\rho_x > \frac{c_{\mathbf{W}}\nu}{\sqrt{1-\eta_s}}$, $\rho_{\mathbf{x}+\mathbf{v}} > 0$.

$\square$

# D   Numerical Experiments Details

The models on images (MNIST and CIFAR10) were trained by minimizing the following regularized empirical risk

$$\min_{\mathbf{W},\mathbf{D}} \frac{1}{m} \sum_{i=1}^{m} \ell\left(y_i, \langle \mathbf{W}, \varphi_{\mathbf{D}}(\mathbf{x}_i) \rangle\right) + \alpha \|\mathbf{I} - \mathbf{D}^T\mathbf{D}\|_F^2 + \beta\|\mathbf{W}\|_F^2, \tag{123}$$

over the training set with $m$ samples. We use the default training/testing split provided in the datasets. The difficulty in this optimization problem resides in computing the derivative of this loss w.r.t. the dictionary $\mathbf{D}$ via the solution of the encoder $\varphi_{\mathbf{D}}(\mathbf{x})$. Our approach relies on using an approximate but differentiable solution for $\varphi_{\mathbf{D}}(\mathbf{x})$: we compute the features (by solving the corresponding Lasso problem) via Fast Iterative Soft Thresholding [Beck and Teboulle, 2009]. This algorithm enjoys a fast convergence rate of $\mathcal{O}(1/T^2)$, and we use $T = 25$ iterations within the optimization problem above.

We found it useful to pre-train the model, $\mathbf{D}$, in an unsupervised manner first. This is done by simply minimizing a regression problem of the form

$$\min_{\mathbf{D}} \frac{1}{m} \sum_{i=1}^{m} \|\mathbf{x}_i - \mathbf{D}\varphi_{\mathbf{D}}(\mathbf{x}_i)\|_2^2.$$

Additionally, when performing the supervised learning stage, if progressively increase the value of $\lambda$ through the iterations until the pre-specified target value (which where set to 0.2 and 0.3 in Figure 2c and Figure 2c, respectively). We employ Adam [Kingma and Ba, 2014] with a mini-batch size of 128, and train for 35 epochs. The dictionary is normalized after each weight-update. All other hyper-parameters are detailed in the accompanying code.

At deployment time, however, it is important that the solution computed by $\varphi_{\mathbf{D}}(\mathbf{x})$ is exact, because the encoder gap $\tau_s$ is defined in terms of these optimality conditions. Therefore, we use FISTA to find the estimated support of the solution, and then compute the exact solution analytically given this support.

All experiments were coded in Python and employing pytorch for GPU acceleration. All other employed packages are detailed in the accompanying code.