[Reviews · NeurIPS 2020]

Review 1

Summary and Contributions: The authors analyze the adversarial robustness of the supervised sparse coding model. They considered a model that involves learning a representation while at the same time giving a precise generalization bound and a robustness certificate. And the analysis is demonstrated by experiments.

Strengths: The theoretic analysis is sufficient. The authors present a generalization bound on the robust risk for the supervised sparse coding model class, and address that the output of the trained model does not change for norm-bounded adversarial perturbations.

Weaknesses: The experimental parts could be augmented.

Correctness: Yes, the claims and method are correct. The empirical methodology is correct.

Clarity: Yes, the paper is well written.

Relation to Prior Work: Yes, it is clearly discussed how this work differs from previous contributions.

Reproducibility: Yes

Additional Feedback: Post-rebuttal update: The authors' response is very nice and detailed, I stay the same score


Review 2

Summary and Contributions: This papers contributes new theoretical results on the adversarial robustness and generalization of a specific hypothesis class - a linear classifier on top of a sparsity-promoting encoder. They leverage the “encoder gap” of the encoder to derive both a generalization bound for adversarial test accuracy and a certificate for robustness. They provide preliminary experimental results on synthetic data and MNIST.

Strengths: This work derives new theoretical results for adversarially robust generalization of a specific hypothesis class, which is significant because adversarially robust generalization is not a well-understood topic. Furthermore, I do not believe this particular model (of a linear classifier + a sparsity-promoting encoder) has been studied extensively for adversarial robustness before, and these results may encourage further work in similar directions.

Weaknesses: The experimental results are somewhat weak. First of all, many of the theoretical claims rely on the encoder gap, and there is only one experimental evaluation of this value a real dataset (MNIST). Even then, MNIST is a very simple dataset, so I am worried about the generalizability of these experiments. Regardless of the value of this encoder gap on other datasets, I would like to see more evaluations (e.g. for something like CIFAR) just because it’s important to understand this quantity to judge if the theoretical contributions will be practically helpful. Furthermore, it is not clear that sparsity-promoting encoders are the right models to be studying, but they do represent a more complex model than linear classifiers.

Correctness: I did not check the math thoroughly, but the parts in the main text and the theoretical results all look reasonable. There are a few assumptions that the theoretical results rely on, but one is reasonable while the second (regarding the encoder gap) is evaluated empirically (though the evaluation should be more extensive). The empirical methodology also appears correct.

Clarity: Overall, the clarity is good. A few points can be clarified. 1. Given the importance of the encoder gap, having a section explaining and analyzing just the encoder gap (e.g. explaining some of the prior work of Mehta and Gray, 2013 as well) might be clearer. 2. In Theorem 4.1, there are many variables that are used that require searching through the text to understand what they are (e.g. the variable b). I prefer the presentation of Theorem 5.1, where all the relevant variables are clearly explained in the Theorem. 3. For Theorem 4.1, it would be good to explain how your result compares to prior theoretical bounds on adversarially robust generalization.

Relation to Prior Work: Yes. Prior work is cited extensively. A bit more background on the encoder gap (Mehta and Gray, 2013) could be explained in this work too.

Reproducibility: Yes

Additional Feedback: Overall, the work achieves new and interesting theoretical results for the model being studied. My main worry is the lack of experimental results on the encoder gap for datasets beyond MNIST, especially given that the size/existence of the encoder gap is crucial to the theoretical results and is an assumption made in the theoretical claims. Thus, I would highly recommend at least evaluating the encoder gap for other (more complex than MNIST) datasets. Many techniques that work well on MNIST may not work on other datasets due to MNIST’s relative simplicity. For example, a network that binarizes pixel values (converts everything below 0.5 to 0, everything above to 1) and then classifies the result is quite adversarially robust, but the same technique will not work for more complex datasets. The paper is borderline for me; I am happy to change my score if the encoder gap can be explained further. ------ After reading all comments from fellow reviewers and the author feedback, I have increased my score from a 5 to a 6 (marginally above the acceptance threshold). The authors provided good explanations of when the encoder gap is non-negative (as long as you make s large enough, it will be), and also justification for why studying sparse coding models is a first step towards studying more general neural networks. I am not familiar enough with the background to be more confident than a 6 in my score. I hope the authors will include the new experimental evaluation and explanations in future revisions (e.g. include explanations in the main body if it makes the paper more clear, and additional experimental info in the Appendix)


Review 3

Summary and Contributions: This paper proposes to apply a sparsity-promoting encoder followed by a linear classifier, which is guaranteed with robustness certificates for classification problems. The proposed method rooted in sparse coding is very intuitive, and the authors also demonstrate the effectiveness of their method on the MNIST dataset.

Strengths: 1). The proposed method enjoys nice theoretical guarantees in terms of robust certificates. The robust certificates derived in this paper are relatively natural in the context of sparse coding (for example, under the encoder gap assumption). 2). In order to provide robust certificates, the proposed method could provide deterministic bounds with much less computation compared with random smoothing. 3). The method could explicitly leverage the potential low-dimensional structure of the data to better defend against adversarial attacks.

Weaknesses: 1). It seems that the method is limited to the linear case as well as the incoherent assumption. It is possible that the data does not satisfy the incoherent assumption or the data could not be represented by linear combinations of columns in D.

Correctness: It is wired that in Figure 2(b), the accuracy is not monotonically decreasing with regard to the adversarial budget.

Clarity: This paper is very well written and easy to follow.

Relation to Prior Work: Yes.

Reproducibility: Yes

Additional Feedback: *** Post-rebuttal *** Thanks for the authors' response. I have carefully read the authors' responses as well as other reviewers' comments. My concerns are addressed in the author response. I think the approach proposed in this paper provides another perspective to study/improve robustness and the method itself is intuitive and theoretically grounded. I think this is a good submission. I will keep my original score and suggest an acceptance. ---------------------------------------------------------------------------------------------------- 1). Although the method is simple and easy to follow, maybe it is better to describe the method in the form of an algorithm and list it in the appendix if there is not enough space in the main body. 2). It is better to provide the code for training and verification for the MNIST experiment.


Review 4

Summary and Contributions: This paper focuses on study the adversarial robustness of the supervised sparse coding model. It provides a uniform generalization bound for the robust risk and a robustness certificate for the learned hypotheses. It can help analyze the representation computed by sparse coding models.

Strengths: This paper provides a detailed study on model robustness of the very specific space coding model. It can be beneficial to the theoretical study within this community.

Weaknesses: The paper studies a relatively narrow problem, which makes it less appealing to wide range of communities. In my perspective, the writing lacks clarity and is hard to follow. Also, it is in inappropriate to mark the text with colors for formal submission.

Correctness: I can not objectively evaluate its correctness, due to my limited knowledge about this area.

Clarity: The paper is writing elaborately. It is written redundantly, making it relatively hard to follow.

Relation to Prior Work: The paper studies a robustness problem for a classic model. Relevant references are adequately presented as preliminaries. However, there seem to be no relative works for parallel comparison regarding contributions.

Reproducibility: Yes

Additional Feedback: I have read the author response and the comments of other reviews. It is claimed that this work provides a nice and convincing theoretical guarantees robust certificates for sparse coding. I would increase my score to 6 and defer to other reviews'comments for final decision. Thank you.

[Author Response · NeurIPS 2020]

We thank the reviewers for their encouraging and instructive comments, and the AC for guiding the review process.

**Encoder Gap and Further Numerical Evidence (R1, R2 and R3)** The encoder gap can be viewed simply as a measure
of maximal energy along any dictionary atom that is not in the support of an input vector. For example, if we assume that
the dictionary forms an orthogonal basis for principal subspace of the data, and the encoder is linear (e.g., as in PCA),
then the expected encoder gap would be $\lambda - (s+1)^{th}$ largest eigenvalue of the covariance matrix $\mathbf{C_{xx}} = \mathbf{E}[\mathbf{xx}^\top]$
(assuming zero-mean data). More generally, it is the $(s+1)^{th}$ entry of the vector $\lambda\mathbf{1} - |\langle \mathbf{D}, \mathbf{x} - \mathbf{D}\varphi_{\mathbf{D}}(\mathbf{x})\rangle|$ (ordered in
increasing manner) as we state on line 123. The formal max-min definition follows the previous work of Mehta and
Gray (2013), and may look a bit too complicated. We will add a remark in line with our comment above.

Note that the assumption on encoder gap is very mild. Intuitively, if a dictionary $\mathbf{D}$ provides quickly decaying
approximation error as a function of the cardinality, then a positive encoder gap exists for some $s$. Importantly, the
cardinality $s$ provides a knob for our results as one can always consider larger $s$ to guarantee that $\tau_s(\mathbf{x}) > 0$, at the
expense of the scaling of our generalization bound (through $\eta_s$). Moreover, one can still induce a larger encoder gap by
increasing the regularization parameter $\lambda$, as demonstrated in Fig. 1 in the paper (and the figures in this document),
which will come at the expense of accuracy (as demonstrated in Fig. 2c and 2d in the manuscript). In this way, our
results guarantee that if one can achieve good accuracy with large encoder gap, then one can generalize robustly. This is
reminiscent of generalization bounds for any margin based predictor (e.g., SVM): If the empirical margin loss is small
and the margin achieved is large, then the hypothesis generalizes well.

Lastly, while our contribution is mostly the-
oretical, we also provide further numerical
evidence for the encoder gap in real scenar-
ios. We trained two models, on SVHN (with
256 atoms) and CIFAR (with 1024 atoms),
with the training procedure described in our
paper, and depict the value of $\tau_s$ (obtained
over a collection of samples), for different
values of $\lambda$. As you can see, one can easily
obtain $\tau_s > 0$ for quite small values of $s$.

**R2:** *It is not clear that sparsity-promoting encoders are the right models to be studying.* Deploying sparse priors in
the learned representations is a first-take at the analysis of non-linear and data-dependent mappings for supervised
models. Ours is the first work to address this. Moreover, *parsimony* (e.g., sparse feature learning) plays an important
role throughout data science and machine learning. Sparsity of learned representation can be ensured by sparsity on the
weight vectors in the second layer of a neural network (the dense first layer can be viewed as learning a dictionary), and
it is indeed common in practice to have an $\ell_1$ penalty on weight matrices. Convolutional structures further promote
sparsity. The challenge in analyzing a two hidden layer neural network (as described above) is that it is unclear what a
good distributional property would be (akin to encoder gap) that will allow a trade-off between robustness and accuracy.

**R2:** *For Theorem 4.1, it would be good to explain how your result compares to prior theoretical bounds on adver-*
*sarially robust generalization.* There is no direct analytical comparison since the nature of work here is quite different:
none of the prior works study the role of margin and stability of the learned representation in enabling a trade-off
between robustness and accuracy. We will add a qualitative comparison and discussion on these in the revised version.

**R3:** *It seems that the method is limited to the linear case as well as the incoherent assumption. It is possible that*
*the data does not satisfy the incoherent assumption or the data could not be represented by linear combinations of*
*columns in D?* Note that the end-to-end map $f(\mathbf{x})$ is *non-linear* in $\mathbf{x}$, as it is the composition of a linear function and a
non-linear representation map. Next, the only assumption we need on the incoherence of the dictionary is that $\eta_s < 1$,
which is mild. The fact that image data can be sparsified by incoherent dictionaries is well-known (e.g. the cornerstone
of JPEG-2000). In practice, this is ensured during training via regularization as described in Equation (7). If you look
at the statement of Theorem 4.1, there is no assumption on $\eta_s$, instead our bounds are in terms of $\eta_s$, so the sample
complexity is expressed directly in terms of this quantity.

**R3:** *It is weird that in Figure 2(b), the accuracy is not monotonically decreasing with regard to the adversarial*
*budget.* Computing adversarial perturbations requires solving a non-convex optimization problem. Since this is
infeasible in general (for non-linear models), one resorts to approximations (such as those based on projected gradient
descent [Madry 2018]). These approximations are not guaranteed to recover the perturbations that maximize the
error, and as a result, the empirical reported accuracy is not necessarily monotonically decreasing. Note though that
fluctuations are very small and within the margin of approximation.

**Experimental details:** We are committed to the reproducibility of our results. All code to reproduce experiments will
be shared and openly available.

[Meta-Review · NeurIPS 2020]

The paper proposes a novel point of view on adversarial attacks, with interesting theoretical results. R2 and R3 have raised a few concerns, and some of them have been addressed by the rebuttal. The area chair agrees with their assessment and recommends to accept the paper. Note that the paper received two poor-quality reviews (R1 and R4), and the decision was mostly based on R2, R3, and the area chair's expertise.